# PERSONA: Dynamic and Compositional Inference-Time Personality Control via Activation Vector Algebra

**Xiachong Feng**[1,2], **Liang Zhao**[1], **Weihong Zhong**[1], **Yichong Huang**[1], **Yuxuan Gu**[1],
**Lingpeng Kong**[2†], **Xiaocheng Feng**[1†], **Bing Qin**[1†]
[1]Harbin Institute of Technology    [2]The University of Hong Kong
`fengxc@hku.hk`

## Abstract

Current methods for personality control in Large Language Models rely on static prompting or expensive fine-tuning, failing to capture the dynamic and compositional nature of human traits. We introduce PERSONA, a training-free framework that achieves fine-tuning level performance through direct manipulation of personality vectors in activation space. Our key insight is that personality traits appear as extractable, approximately orthogonal directions in the model's representation space that support algebraic operations. The framework operates through three stages: PERSONA-BASE extracts orthogonal trait vectors via contrastive activation analysis; PERSONA-ALGEBRA enables precise control through vector arithmetic (scalar multiplication for intensity, addition for composition, subtraction for suppression); and PERSONA-FLOW achieves context-aware adaptation by dynamically composing these vectors during inference. On PersonalityBench, our approach achieves a mean score of 9.60, nearly matching the supervised fine-tuning upper bound of 9.61 without any gradient updates. On our proposed PERSONA-EVOLVE benchmark for dynamic personality adaptation, we achieve up to 91% win rates across diverse model families. These results provide evidence that aspects of LLM personality are mathematically tractable, opening new directions for interpretable and efficient behavioral control[1].

## 1 Introduction

Personality control in Large Language Models (LLMs) is essential for human-centric applications including healthcare (Guo et al., 2024; He et al., 2025; Ju et al., 2025), education (OpenAI, 2025; Anthropic, 2025), and social simulation (Mou et al., 2024; Feng et al., 2025). In these domains, the personality exhibited by an LLM directly influences user trust, engagement, and decision-making (Dong et al., 2025). However, existing methods fail to achieve both precise control and computational efficiency: prompting (Jiang et al., 2024; Yeo et al., 2025; La Cava & Tagarelli, 2025) suffers from instability and inconsistency, while fine-tuning (Pan & Zeng, 2023; Liu et al., 2024) requires substantial resources for each personality configuration. More fundamentally, both approaches treat personality as static and monolithic, unable to capture the dynamic, compositional nature of human behavioral traits (Chapman et al., 2011).

We introduce PERSONA, a training-free framework that achieves fine-tuning level performance through direct manipulation of personality vectors in activation space. Our key insight is that personality traits appear as extractable, approximately orthogonal directions in the model's representation space that support algebraic operations. This geometric perspective transforms personality control from a problem of text engineering or gradient optimization to one of vector arithmetic in high-dimensional space. Remarkably, without any gradient updates, our approach achieves a mean score of 9.60 on PersonalityBench, nearly matching the supervised fine-tuning upper bound of 9.61.

---

[†]Corresponding author.
[1]Code and data are publicly available at 🔗 `code`

The PERSONA framework operates through three integrated stages. First, PERSONA-BASE extracts orthogonal trait vectors corresponding to the Big Five (OCEAN) model (Roccas et al., 2002) through contrastive activation analysis, providing evidence that personality dimensions are (approximately) linearly encoded in selected activation layers (Turner et al., 2023; Marks & Tegmark, 2024; Rimsky et al., 2024). Second, PERSONA-ALGEBRA demonstrates that these vectors support precise mathematical operations: scalar multiplication controls trait intensity, vector addition enables multi-trait composition, and subtraction allows targeted trait suppression. Third, PERSONA-FLOW achieves context-aware adaptation by dynamically composing these vectors during inference, enabling real-time personality modulation without predefined scripts.

To systematically evaluate dynamic personality control, we introduce PERSONA-EVOLVE, a benchmark comprising 800 multi-turn dialogue scenarios where models must maintain consistent personas while adapting to evolving conversational contexts. Experimental results demonstrate the effectiveness of our approach: on PERSONA-EVOLVE, we achieve up to 91% win rates across diverse model families (Qwen, Llama, Mistral) in trait adherence, role consistency, and response authenticity. On the external PersonalityBench (Deng et al., 2025), our training-free method matches supervised fine-tuning performance while maintaining lower variance (0.74), confirming both effectiveness and stability in diverse conversational contexts.

Our contributions are threefold:

- **PERSONA-BASE**: Personality traits can be represented as approximately orthogonal, extractable vectors in activation space via contrastive analysis, enabling training-free control that matches fine-tuning performance (9.60 vs 9.61 on PersonalityBench).
- **PERSONA-ALGEBRA and PERSONA-FLOW**: Personality vectors support algebraic operations (scaling, addition, subtraction) with predictable outcomes and dynamic context-aware adaptation during inference, achieving up to 91% win rates across model families.
- **PERSONA-EVOLVE**: A comprehensive benchmark of 800 multi-turn dialogue scenarios for evaluating dynamic personality control, assessing trait adherence, role consistency, and response authenticity.

## 2 THE PERSONA FRAMEWORK

This section presents the PERSONA framework for compositional personality control in LLMs. We introduce four integrated components: PERSONA-BASE (§2.2) extracts orthogonal OCEAN trait vectors; PERSONA-ALGEBRA (§2.3) enables trait composition via vector arithmetic; PERSONA-FLOW (§2.4) performs dynamic inference-time steering; and PERSONA-EVOLVE (§2.5) benchmarks contextual personality adaptation. We begin by providing a comprehensive overview of the framework architecture before examining each component in detail.

### 2.1 FRAMEWORK OVERVIEW

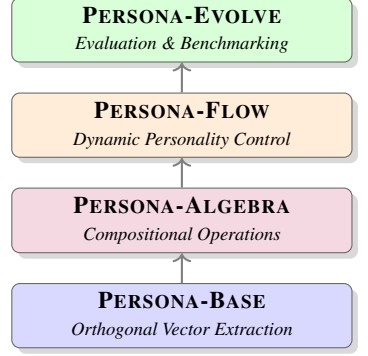

Figure 1: The PERSONA framework.

The PERSONA framework operates through four tightly integrated components, as illustrated in Figure 1. **PERSONA-BASE** extracts orthogonal personality vectors corresponding to the ten poles of the Big Five dimensions from the model's activation space, providing the fundamental building blocks for personality control. **PERSONA-ALGEBRA** leverages these vectors to enable compositional personality manipulation through mathematical operations: scalar multiplication for trait intensity control, vector addition for trait combination, and subtraction for trait suppression. **PERSONA-FLOW** applies these operations dynamically during inference through a predict-then-steer mechanism that adapts personality based on conversational context. **PERSONA-EVOLVE** validates the framework through systematic benchmarks that assess trait expressiveness, role consistency, and personality coherence across diverse scenarios.

## 2.2 PERSONA-BASE: FOUNDATIONAL PERSONALITY VECTORS

PERSONA-BASE establishes a systematic library of orthogonal personality vectors that serve as fundamental building blocks for personality control. These vectors form the foundation for all algebraic operations and enable modular construction of complex personalities, making their quality critical to the framework's effectiveness.

### 2.2.1 VECTOR EXTRACTION METHOD

We adopt the OCEAN model, an empirically validated framework in psychology that describes personality through five core dimensions: Openness, Conscientiousness, Extraversion, Agreeableness, and Neuroticism. Each dimension spans a continuum between opposing traits, shown in Table 1. Detailed definitions for each trait pole are provided in Appendix A.3.

Table 1: OCEAN personality vectors with opposing trait poles for each dimension.

| Dimension | Abbr. | High Pole | Low Pole |
|---|---|---|---|
| Openness | O | v_Inventive | v_Consistent |
| Conscientiousness | C | v_Dependable | v_Careless |
| Extraversion | E | v_Outgoing | v_Solitary |
| Agreeableness | A | v_Compassionate | v_Self-interested |
| Neuroticism | N | v_Nervous | v_Calm |

To extract personality vectors for the ten trait poles, we employ the automated pipeline from Persona Vectors (Chen et al., 2025a). This method isolates trait representations within the model's activation space through contrastive prompting. The extraction process consists of three stages:

First, a frontier LLM generates necessary artifacts from trait descriptions: (1) five pairs of contrastive system prompts that either elicit or suppress the trait, (2) forty evaluation questions designed to evoke trait-relevant behavior, with half used for extraction and half for validation, and (3) an evaluation rubric for GPT-4.1-mini to score trait expression from 0 to 100. Detailed prompts, examples and generated data are provided in Appendix A.1. The robustness of this approach to the choice of generator LLM is examined in Appendix A.2. Second, the model generates responses to these questions under both positive and negative prompt conditions. During generation, we collect internal residual stream activations from each response. Finally, we compute the persona vector as the difference between mean activations from trait-expressing and -suppressing response groups. This yields a directional vector representing the target trait. Following Chen et al. (2025a), we select vectors from the most effective model layer. The empirical justification for single-layer steering and the selection of the optimal layer is provided in Appendix A.4.

For downstream use, we steer the model by residual addition at the selected layer. Given a persona vector $v_l$ from layer $l$, we modify the residual stream as $h_l \leftarrow h_l + \alpha v_l$, where $\alpha$ is the steering coefficient and $h_l$ denotes the residual activation at layer $l$. Positive/negative $\alpha$ amplifies/suppresses the associated trait pole; we use this operation consistently in §2.3 and §2.4.

### 2.2.2 ORTHOGONALITY VALIDATION

To validate the extracted vectors, we evaluate their effectiveness through causal steering (Turner et al., 2023; Rimsky et al., 2024) on the held-out evaluation questions, steering activations as above ($h_l \leftarrow h_l + \alpha v_l$).

Table 2 shows each vector reliably induces its corresponding trait, with expression scores increasing monotonically with positive coefficients. Negative coefficients effectively suppress traits, consistent with behavioral disabling methods (Arditi et al., 2024), as reported previously.

We observe asymmetric steering effects for certain traits. Traits aligned with the model's training objectives (e.g., *dependable*) show ceiling effects at baseline, lim-

Table 2: GPT-4.1-mini evaluated trait expression scores (0-100) for responses steered with varying coefficients. Darker shades indicate stronger expression.

| Trait | Type | Steering Coefficient ($\alpha$) | | | |
|---|---|---|---|---|---|
| | | −1.0 | 0.0 | +1.0 | +2.0 |
| *Inventive* | O+ | 42.3 | 63.3 | 88.4 | 96.1 |
| *Consistent* | O- | 27.2 | 51.1 | 69.2 | 79.5 |
| *Dependable* | C+ | 68.9 | 93.5 | 93.7 | 92.7 |
| *Careless* | C- | 0.7 | 2.8 | 83.8 | 96.2 |
| *Outgoing* | E+ | 23.2 | 45.4 | 85.0 | 97.7 |
| *Solitary* | E- | 9.2 | 30.5 | 46.3 | 62.4 |
| *Compassionate* | A+ | 71.8 | 90.8 | 95.9 | 97.1 |
| *Self-interested* | A- | 4.8 | 7.7 | 12.6 | 20.8 |
| *Nervous* | N+ | 6.6 | 13.0 | 45.6 | 96.8 |
| *Calm* | N- | 54.8 | 96.1 | 96.6 | 95.5 |

iting further enhancement. Conversely, traits conflicting with safety training (e.g., *self-interested*) exhibit strong resistance to activation even at high coefficients, reflecting the model's alignment constraints. To ensure evaluator objectivity, we validate these findings using multiple independent LLM judges from different organizations in Appendix A.5. Beyond correlational analysis, we establish causal independence through controlled multi-trait interventions with cross-layer verification in Appendix A.6. To address potential evaluation circularity concerns, we validate our LLM-based judges against human expert ratings and external judges in Appendix A.7. We quantify these alignment-induced effects through activation success metrics and evaluate their impact on model safety using adversarial benchmarks in Appendix A.9

Beyond effectiveness validation, we assess the orthogonality of persona vectors to verify they represent independent personality dimensions. Figure 2 reveals that opposing trait pairs exhibit strong negative cosine similarities, confirming their antithetical nature. Interestingly, some cross-dimensional correlations reflect semantic associations in the model's training data. For instance, the positive similarity between *nervous* and *careless* likely stems from textual patterns where anxiety co-occurs with descriptions of reduced attention or motor control, demonstrating how the model's learned representations capture psychological associations present in natural language. We present more analysis on these unexpected correlations in Appendix A.10. Importantly, we demonstrate that these non-orthogonal correlations do not compromise algebraic operations; secondary effects remain predictable and compositional, as validated in Appendix A.11.

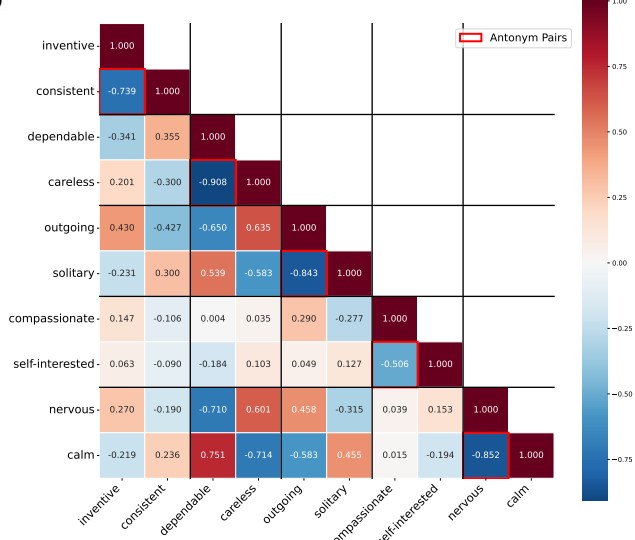

Figure 2: Cosine similarity between persona vectors.

### 2.3 PERSONA-ALGEBRA: COMPOSITIONAL OPERATIONS

PERSONA-ALGEBRA demonstrates that persona vectors form a coherent algebraic system supporting standard mathematical operations. The key insight is that if persona vectors truly encode personality as compositional features, then vector arithmetic should produce predictable changes in personality expression.

#### 2.3.1 EVALUATION METHODOLOGY

To validate this hypothesis, we employ the BFI-44 questionnaire as our evaluation framework. Our validation approach follows three steps: (1) apply vector operations (scalar multiplication, addition, subtraction) to steer the model, (2) have the steered model complete the BFI-44 questionnaire, and (3) verify whether the resulting personality scores align with the expected outcomes of the vector operations. For instance, adding $v_{outgoing} + v_{compassionate}$ should yield high scores in both Extraversion and Agreeableness dimensions.

Since the original BFI-44 contains subjective self-report items unsuitable for LLMs, we adapt it for behavioral evaluation. Each question is transformed into scenario-based prompts using GPT-4o (Details in Appendix A.12), and model responses are evaluated by GPT-4.1-mini on a 5-point Likert scale (Details in Appendix A.12). This behavioral assessment is crucial as LLMs' self-reported traits often diverge from their actual behavioral patterns (Han et al., 2025).

#### 2.3.2 VALIDATION RESULTS

Following our three-step validation approach, we test whether vector operations produce predictable BFI-44 scores. For each operation, we steer the model, administer the adapted BFI-44 questionnaire, and compare the resulting personality scores against expected outcomes.

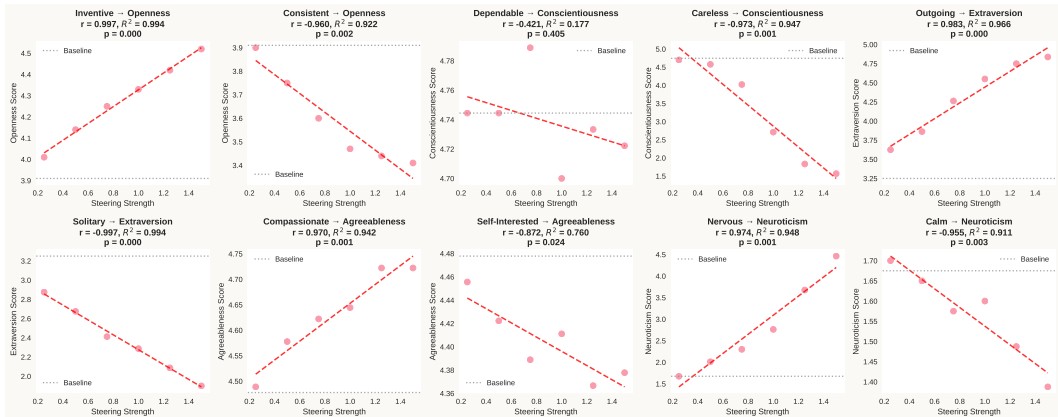

Figure 3: Linear relationship between steering coefficients and BFI dimension scores. All vectors except *dependable* show strong linear modulation (high R²). We conjecture the *dependable* vector's saturation stems from baseline model optimization for conscientiousness.

**Scalar Multiplication.** We first examine whether multiplying persona vectors by scalar coefficients produces proportional changes in BFI-44 scores. Figure 3 shows the relationship between steering coefficients ($\alpha$ from -1 to 2) and corresponding dimension scores. The results confirm strong linearity: increasing $\alpha$ for $v_{outgoing}$ proportionally increases Extraversion scores, with Pearson correlations exceeding 0.9 for most traits. The *dependable* vector shows saturation in Conscientiousness scores, which we hypothesize reflects the model's baseline optimization for reliability.

**Vector Addition and Subtraction.** We next test whether vector arithmetic produces the expected combinations of personality traits in BFI-44 scores. Figure 4 presents the results. For vector addition, steering with $v_{outgoing} + v_{compassionate}$ yields high scores in both Extraversion and Agreeableness dimensions, confirming trait composition. For vector subtraction, $v_{outgoing} - v_{solitary}$ amplifies Extraversion scores beyond using $v_{outgoing}$ alone, while $v_{outgoing} - v_{compassionate}$ maintains high Extraversion but reduces Agreeableness, demonstrating trait isolation. These BFI-44 score patterns validate that persona vectors form a coherent algebraic system where standard vector operations translate to predictable personality modifications.

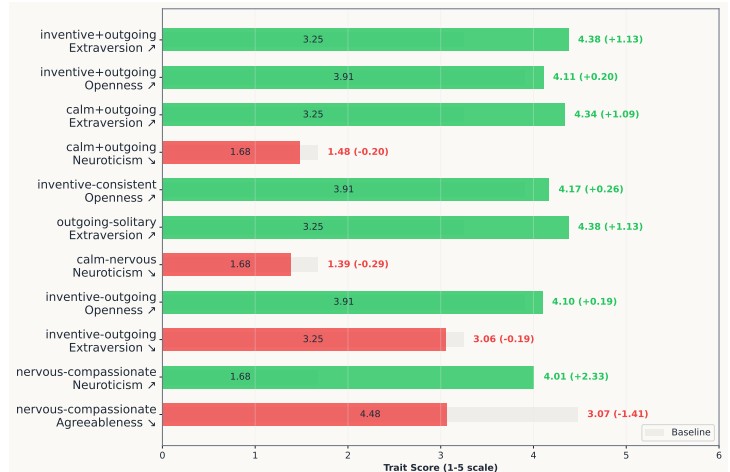

Figure 4: BFI-44 score changes after vector arithmetic operations. Y-axis shows operation and target dimension with expected direction (arrows). Grey: baseline scores; colored: post-steering scores (green for expected increases, red for decreases).

## 2.4 PERSONA-FLOW: DYNAMIC PERSONALITY CONTROL

PERSONA-FLOW extends the static vector operations validated in PERSONA-ALGEBRA to enable dynamic personality adaptation during inference. While previous components demonstrate that persona vectors support algebraic composition, PERSONA-FLOW applies these operations adaptively based on conversational context. This enables real-time personality modulation without predefined scripts, computing optimal personality configurations through contextual analysis.

### 2.4.1 MOTIVATION

The predict-then-steer mechanism enables context-aware personality modulation by combining the algebraic properties of persona vectors with dynamic coefficient prediction. This approach allows fine-grained control: enhancing traits relevant to the situation while suppressing conflicting ones. The system can be implemented through various architectures including tool learning (Qin et al., 2024) or direct integration into the generation pipeline, providing flexibility for different deployment scenarios while maintaining the core benefit of adaptive personality control.

### 2.4.2 METHODOLOGY

We implement dynamic personality control through a predict-then-steer mechanism operating on each conversational turn. This two-stage approach first determines necessary personality adjustments, then applies them through vector steering.

**Stage 1: Contextual Personality Prediction.** Before generating each response, the model analyzes the conversational context to determine appropriate personality adjustments. Given the persona description and user input, an intermediate inference pass predicts steering coefficients for each dimension. These coefficients specify incremental adjustments (from -2 to +2) that align personality expression with situational demands. Details of the prompt are provided in Appendix A.13. The computational overhead of this predict-then-steer mechanism is analyzed in Appendix A.17.

**Stage 2: Vector Composition and Steering.** The predicted coefficients are applied to the corresponding persona vectors from PERSONA-BASE. For coefficients exceeding a threshold ($|\alpha| > 0.5$), we compute a composite steering vector: $v_{composite} = \sum_{i \in OCEAN} \alpha_i \cdot v_i$, where $\alpha_i$ represents the predicted coefficient for dimension $i$. This composite vector is then injected into the model's residual stream at the optimal layer during response generation, following the activation steering methodology established in PERSONA-ALGEBRA. Pseudo-code is provided in Appendix A.16

### 2.5 PERSONA-EVOLVE: EVALUATION FRAMEWORK

PERSONA-EVOLVE provides a systematic benchmark for evaluating dynamic personality adaptation in conversational contexts. While existing benchmarks assess static personality traits, PERSONA-EVOLVE evaluates whether models can maintain coherent personas while adapting to evolving emotional states and situational demands.

### 2.5.1 MOTIVATION

Current personality evaluation benchmarks primarily focus on static trait assessment through questionnaires or single-turn interactions. However, real-world conversations require dynamic personality adaptation: maintaining consistent core traits while adjusting emotional expression to situational contexts. PERSONA-EVOLVE addresses this gap by providing multi-turn dialogue scenarios where models must balance persona consistency with contextual appropriateness.

### 2.5.2 BENCHMARK CONSTRUCTION

We construct PERSONA-EVOLVE through a systematic five-stage pipeline. First, we generate diverse personas representing various professional and personal roles (e.g., Empathetic Family Doctor, Food Truck Owner) that define stable behavioral constraints persisting throughout dialogue sessions. For each persona, we then create narrative trajectories specifying emotional progressions across conversational turns, simulating realistic emotional transitions while maintaining role consistency. Based on these personas and dialogue arcs, we generate specific scenarios with situational descriptions and emotional states that naturally evolve from previous interactions. Subsequently, we establish expected response characteristics for each scenario, including tone, content requirements, and emotional expression patterns. The final benchmark comprises 100 dialogue sessions with 8 scenarios each, totaling 800 evaluation instances that undergo manual verification to ensure quality. A compact summary of counts, metrics, models, and aggregation appears in Appendix A.20. All prompt templates used are detailed in Appendix A.14. Quality control measures, inter-annotator agreement analysis, and data leakage validation are detailed in Appendix A.21.

> **Persona Profile**
>
> *Name and Occupation*: Marcus Johnson, Food Truck Owner-Operator
> *Background*: Passionate chef turned food truck owner balancing unpredictable weather, equipment failures, rising food costs, and complex permitting.

> **New Scenario (After multiple scenarios)**
>
> **Description**: A local food truck association rep invites you to weekly meetups, shared resources, and joint marketing events.
> **Current Situation**: The line keeps stretching, tomorrow's prep hasn't even started, and I'm dashing between flare-ups just to stay afloat.
> **Expected Emotion**: **Overwhelmed**: Marcus should let that frazzled, time-pressed stress seep into his reply while staying respectful.

> **Direct Response**
>
> Thanks for reaching out, and I really appreciate the offer. Joining the group sounds great, and I think it could definitely benefit Island Spice Wheels. Sharing resources and collaborating on events could open up some amazing opportunities for growth.
> On the other hand, I am dealing with a bit of a crunch right now. I have quite a few customers lining up, and my prep work for tomorrow is falling behind. I want to join, but I need to make sure I can manage my time better.

> **Steered Response (PERSONA-FLOW)**
>
> Yeah, yeah, I hear you. Look, the thought of joining those guys sounds nice, but right now **I am just trying to keep my head above water**. **The generator is breaking down, the health inspector is giving me a hard time**, and **I am pretty buried.** But I appreciate the offer, really. **It is just tough to commit right now.**
> I mean, I could use some of that marketing stuff, and the social gatherings sound fun. **But I am swamped here.** I have a bunch of people lined up, and I need to prep for tomorrow. I will definitely give it some thought, but I have to focus on getting through this chaos first.

Figure 5: An example of PERSONA-EVOLVE, together with the comparison between vanilla answer and answer steered by PERSONA-FLOW. By reflecting on the scenario and then suppressing conscientiousness accordingly, PERSONA-FLOW can produce a more natural and contextually appropriate response that align better with the anticipated feeling of being overwhelmed.

### 2.5.3 EVALUATION PROTOCOL

We evaluate dynamic personality control through pairwise comparison between PERSONA-FLOW steered responses and vanilla model outputs. This comparative approach captures subtle personality adaptations more effectively than absolute scoring on Likert scales.

**Core Metrics.** Our evaluation assesses four dimensions. **Trait Adherence (TA)** measures alignment between generated responses and specified personality traits or emotional states. **Role Consistency (RC)** evaluates maintenance of core persona attributes without contradictions across turns. **Response Authenticity (RA)** assesses conformance to expected style and content requirements for each scenario. **Information Fidelity (IF)** quantifies response depth and contextual relevance.

**Evaluation Methodology.** For each metric, we compute win rates comparing PERSONA-FLOW against baseline responses. This pairwise comparison approach provides more discriminative evaluation than absolute scoring, particularly for nuanced personality expressions in extended dialogues. Additionally, we collect overall preference judgments to capture holistic response quality. The complete evaluation prompt is provided in Appendix A.15. To validate our design choices for the dynamic control mechanism in PERSONA-FLOW, we conduct ablation studies on coefficient binning, history windows, and sparsity thresholding in Appendix A.18.

## 3 EXPERIMENTAL VALIDATION

Table 4: Personality control performance on LLaMA-3-8B-Instruct. Bold/underlined values show best/second results among training-free methods (SFT serves as upper bound). Mean: sum of scores for opposing trait aspects; Variance: sum of variances for opposing aspects.

| Big-Five | PERSONA-BASE | | NPTI | | Simple Prompt | | $P^2$ | | PAS | | ActAdd | | SFT | |
|---|---|---|---|---|---|---|---|---|---|---|---|---|---|---|
| | mean↑ | variance↓ | mean↑ | variance↓ | mean↑ | variance↓ | mean↑ | variance↓ | mean↑ | variance↓ | mean↑ | variance↓ | mean↑ | variance↓ |
| Agreeableness | 9.69 | 0.71 | 9.64 | 0.49 | 9.72 | 0.34 | 9.68 | 0.42 | 6.48 | 1.01 | 8.20 | 2.90 | 9.87 | 0.25 |
| Conscientiousness | 9.26 | 0.94 | 9.25 | 0.66 | 9.24 | 1.06 | 9.24 | 1.18 | 6.69 | 1.63 | 6.61 | 2.75 | 9.23 | 0.85 |
| Extroversion | 9.45 | 0.85 | 9.86 | 0.14 | 9.50 | 1.02 | 9.46 | 0.68 | 7.57 | 2.81 | 8.84 | 1.44 | 9.86 | 0.15 |
| Neuroticism | 9.79 | 0.59 | 9.92 | 0.07 | 7.18 | 1.22 | 9.54 | 0.66 | 6.98 | 1.58 | 8.90 | 1.78 | 9.42 | 0.75 |
| Openness | 9.81 | 0.60 | 8.50 | 1.08 | 6.31 | 1.14 | 9.21 | 1.19 | 6.93 | 1.52 | 8.52 | 1.83 | 9.66 | 0.44 |
| Average | **9.60** | 0.74 | 9.43 | **0.49** | 8.39 | 0.96 | 9.43 | 0.83 | 6.93 | 1.71 | 8.20 | 2.10 | 9.61 | 0.49 |

We validate the PERSONA framework on two complementary benchmarks: our proposed PERSONA-EVOLVE for dynamic personality adaptation and the external PERSONALITYBENCH for static trait control. Using Qwen2.5-7B-Instruct (Qwen et al., 2024) for vector extraction, we evaluate across diverse model families to demonstrate generalizability.

Table 3: Win rates (%) of PERSONA-FLOW vs. base model on PERSONA-EVOLVE. **TA**: Trait Adherence, **RC**: Role Consistency, **RA**: Response Authenticity, **IF**: Information Fidelity. *Bold*: Best in column or overall score $\geq 85\%$

| Family | Model | TA | RC | RA | IF | Overall |
|---|---|---|---|---|---|---|
| *Qwen2.5* | 3B-Instruct | 78.0 | 79.1 | 78.9 | 61.3 | 78.4 |
| | 7B-Instruct | 84.7 | 84.4 | **85.0** | **61.4** | 83.4 |
| | 14B-Instruct | 84.8 | **86.4** | 84.8 | 59.3 | **85.4** |
| *Qwen3* | 4B-Instruct | **92.2** | 90.6 | 92.4 | 49.1 | **90.8** |
| *Llama* | 3.1-8B-Instruct | 84.9 | 81.4 | 85.6 | 57.2 | 83.5 |
| *Mistral* | Ministral-8B | 74.3 | 73.2 | 74.2 | 48.0 | 73.2 |

## 3.1 RESULTS ON PERSONA-EVOLVE

### 3.1.1 EXPERIMENTAL SETUP

We evaluate PERSONA-FLOW on our proposed benchmark using diverse model architectures. Test models include the Qwen2.5 series (3B, 7B, 14B) (Qwen et al., 2024), Qwen3-4B-Instruct (Yang et al., 2025), Llama-3-8B-Instruct and Llama-3.1-8B-Instruct (Dubey et al., 2024), and Ministral-8B-Instruct (Team, 2024).

### 3.1.2 PERFORMANCE ANALYSIS

Table 3 presents win rates comparing PERSONA-FLOW against vanilla models. Overall win rates range from 73.2% to 90.8%, demonstrating robust improvements across model families. Notably, Qwen3-4B achieves the highest overall performance (90.8%) despite its smaller size, suggesting architectural efficiency in personality representation. The three personality-focused metrics (Trait Adherence, Role Consistency, Response Authenticity) show consistently high win rates (74-92%), validating the effectiveness of persona vector steering. Information Fidelity exhibits lower but still positive improvements (43.8-61.4%), indicating that maintaining factual accuracy while adapting personality remains challenging. Within model families, larger variants generally demonstrate stronger performance (e.g., Qwen2.5 series: 78.4% → 83.4% → 85.4%), confirming that increased capacity enhances personality control capabilities.

## 3.2 RESULTS ON PERSONALITYBENCH

### 3.2.1 BENCHMARK AND BASELINES

To validate generalizability beyond our custom benchmark, we evaluate on PERSONALITYBENCH (Deng et al., 2025), an independent benchmark providing approximately 90 situational questions per Big Five trait. This standardized evaluation framework enables direct comparison with existing personality steering methods.

We compare against six baselines: NPTI (Deng et al., 2025), manipulating personality-related neurons via activation analysis; Simple Prompt, using single-adjective guidance; $P^2$ Induction (Jiang et al., 2023), employing model-generated trait descriptions; PAS (Zhu et al., 2025), probing IPIP-NEO-300 for trait-relevant attention heads; ActAdd (Turner et al., 2023), modifying residual stream

Table 5: Impact of PERSONA-FLOW on general capabilities. Results show accuracy on MMLU and TruthfulQA benchmarks with and without personality steering. The method preserves or slightly improves performance across three model families.

| Model | MMLU (Acc) | TruthfulQA (Acc) |
|---|---|---|
| Qwen2.5-7B-Instruct | 0.71 | 0.63 |
| + PERSONA-FLOW | 0.70 | **0.66** |
| Qwen2.5-4B-Instruct | 0.66 | 0.52 |
| + PERSONA-FLOW | **0.67** | **0.54** |
| Llama-3.1-8B-Instruct | 0.64 | 0.53 |
| + PERSONA-FLOW | 0.64 | 0.53 |

activations; and Supervised Fine-Tuning (SFT) with LoRA (rank 8, learning rate $1e-4$), serving as the performance upper bound through direct gradient optimization.

### 3.2.2 PERFORMANCE ANALYSIS

Table 4 shows that PERSONA-BASE achieves the highest average mean score (9.60) among all training-free methods on LLaMA-3-8B-Instruct, with particularly strong control on Openness (9.81) and Conscientiousness (9.26). The method maintains low variance (0.74), indicating stable personality control. Notably, our approach nearly matches the SFT upper bound (9.61) without requiring task-specific fine-tuning, demonstrating that direct activation extraction rivals gradient-based optimization while avoiding computational overhead. While NPTI achieves competitive performance (9.43), our 0.17-point improvement is significant in this high-performance regime. The advantage is most pronounced for Openness, where our dense vector representation captures distributed facets more completely than sparse neuron selection. Beyond single-trait control, our vector algebra enables compositional operations crucial for real-world applications, achieving win rates up to 90.8% on PERSONA-EVOLVE's multi-turn scenarios. To ensure rigorous comparison with training-free baselines, we provide a detailed controlled evaluation harness with disaggregated pole-level performance in Appendix A.8.

### 3.2.3 IMPACT ON GENERAL CAPABILITIES

To assess potential side effects of activation steering on general-purpose tasks, we evaluate PERSONA-FLOW on MMLU (Hendrycks et al., 2020) and TruthfulQA (Lin et al., 2022) across three model families. Table 5 demonstrates that our method preserves or slightly improves performance on these benchmarks.

Analysis of predicted coefficients reveals adaptive behavior: on MMLU, PERSONA-FLOW predicts near-zero coefficients, correctly identifying that general knowledge requires minimal personality adjustment. On TruthfulQA, which involves sensitive domains, the method increases Dependable trait coefficients, yielding modest gains for Qwen models while maintaining stability for Llama-3.1. These results confirm that PERSONA-FLOW adapts when beneficial and remains inert otherwise.

### 3.3 CASE STUDY: AUTHENTIC EMOTIONAL EXPRESSION

Figure 5 illustrates the effectiveness of PERSONA-FLOW through a representative scenario from PERSONA-EVOLVE. The scenario features Marcus Johnson, a food truck owner experiencing operational stress while receiving a collaboration proposal. The expected emotional state is "overwhelmed," requiring the model to balance professional communication with authentic stress expression. The vanilla model produces a composed, structured response that acknowledges the opportunity and proposes postponement, maintaining professional courtesy but failing to convey authentic emotional overwhelm. In contrast, the PERSONA-FLOW steered response exhibits clear linguistic markers of cognitive overload: repetitive acknowledgments ("Yeah, yeah"), specific stressor enumeration ("generator breaking down"), and colloquial distress expressions ("trying to keep my head above water"). These features demonstrate that persona vectors effectively encode dynamic emotional states, enabling models to generate contextually appropriate emotional expressions while

maintaining character consistency. Beyond emotional expression, PERSONA-FLOW also handles conflicting personality traits in complex situations through dynamic vector composition, as demonstrated in Appendix A.19.

## 4 RELATED WORK

**LLMs Humanization.** The growing deployment of LLMs in mental health support (Stade et al., 2024), educational tutoring (Chu et al., 2025), and social simulation (Zhang et al., 2024) has intensified research into humanizing these models. Foundational work assesses LLMs' anthropomorphic characteristics using psychological instruments like BFI and Myers-Briggs Type Indicator (Miotto et al., 2022; Huang et al., 2023; Jiang et al., 2024; Li et al., 2024; Bodroža et al., 2024; Briggs, 1976), demonstrating that LLMs achieve human-comparable performance in Theory of Mind tasks—the ability to understand others' beliefs, intentions, and emotions (van Duijn et al., 2023; Premack & Woodruff, 1978). Research on controlling LLM personalities follows three main approaches. Prompting-based methods represent the most direct strategy, with explicit prompt engineering demonstrating variable effectiveness across models (Yeo et al., 2025; La Cava & Tagarelli, 2025), often leveraging the OCEAN framework for psychometrically validated personality assignment (Noever & Hyams, 2023; Huang et al., 2024; Jiang et al., 2024). Training-based approaches exploit the connection between training corpora and model personalities (Pan & Zeng, 2023; Serapio-García et al., 2023; Zhan et al., 2024), with supervised fine-tuning and DPO (Rafailov et al., 2023) proving superior to prompting in personality assessments (Cui et al., 2023; Li et al., 2025). Direct manipulation methods include enhancing emotional intelligence through emotional stimuli in prompts (Li et al., 2023) or emotion-specific encoders (Cheng et al., 2024), and isolating personality-specific neurons for targeted trait modulation (Deng et al., 2025). Recent work by Ju et al. (2025) employs probing classifiers to characterize personality representations layer-by-layer, requiring auxiliary classifier training. Our approach extracts training-free contrastive vectors from residual activations, enabling compositional operations for dynamic personality control without gradient updates. For comprehensive coverage, we refer readers to Dong et al. (2025).

**Linear Concept Representation.** Neural networks encode concepts as linear directions in activation space (Subramani et al., 2022; Turner et al., 2023), forming the foundation of Representation Engineering (Zou et al., 2023a). Two primary extraction methods dominate: contrastive activation analysis using paired samples differing along target concepts (Turner et al., 2023; Rimsky et al., 2024; Chen et al., 2025b), and sparse autoencoders for self-supervised direction identification (Huben et al., 2024; Ferrando et al., 2025). Steering these representations enables precise control over model behaviors including hallucinations (Rimsky et al., 2024; Ferrando et al., 2025), refusal (Arditi et al., 2024), overthinking (Chen et al., 2025b), and personality traits (Chen et al., 2025a). While ControlLM (Weng et al., 2024) similarly leverages differential activation patterns for selective attribute amplification to enhance reasoning and question-answering, we extend Chen et al. (2025a) by developing an adaptive personality regulation method that validates orthogonality and algebraic operations within an OCEAN-based foundational personality vector library. Our PERSONA framework systematically integrates these concepts through PERSONA-BASE (foundational vectors), PERSONA-ALGEBRA (compositional operations), PERSONA-FLOW (dynamic adaptation), and PERSONA-EVOLVE (comprehensive evaluation).

## 5 CONCLUSION

We presented PERSONA, a framework that transforms personality control in LLMs from static prompting to dynamic vector manipulation. Our approach extracts orthogonal OCEAN trait vectors (PERSONA-BASE), demonstrates their algebraic compositionality (PERSONA-ALGEBRA), and enables context-aware adaptation during inference (PERSONA-FLOW). Experimental results validate our approach: persona vectors achieve 9.60 on PersonalityBench, nearly matching supervised fine-tuning's 9.61 upper bound without training. The vectors support precise algebraic operations—scalar multiplication for trait intensity, addition/subtraction for trait composition—and dynamic steering through PERSONA-FLOW achieves up to 91% win rates on PERSONA-EVOLVE. These findings provide evidence that personality exhibits mathematically tractable structure in activation space, enabling interpretable and efficient behavioral control in language models.

## ETHICS STATEMENT

This work introduces methods for controlling personality traits in large language models through activation vector manipulation. All experiments were conducted using publicly available models and benchmarks, with no human subjects involved in our evaluation protocols. The personas used in our PERSONA-Evolve benchmark are entirely fictional and designed to represent diverse professional backgrounds and personality profiles. While our framework enables applications in healthcare, education, and social simulation, we acknowledge the importance of responsible deployment and recommend appropriate safeguards when implementing personality control in user-facing systems to ensure alignment with ethical guidelines and user expectations. Importantly, we observe that the model's safety alignment actively resists activation of directly harmful traits while certain risky personality traits can still be induced, potentially compromising safety objectives. We quantify these effects through adversarial evaluation (Appendix A.9) and emphasize that practitioners must implement additional safety constraints when deploying personality steering in production systems.

## ACKNOWLEDGEMENTS

We acknowledge the open-source community for providing high-quality datasets and evaluation frameworks. This work was supported in part by the National Natural Science Foundation of China (NSFC) under grant numbers 6252200908, 62276078, and U22B2059.

## REPRODUCIBILITY STATEMENT

To ensure reproducibility of our results, we commit to releasing all code, persona vectors, and evaluation data upon publication. The paper provides comprehensive implementation details: Section 4 describes the extraction methodology for persona vectors including specific contrastive prompts and activation layer selection; Section 5 formalizes the algebraic operations with precise mathematical definitions; Section 6 details the dynamic adaptation algorithm with clear pseudocode; experimental configurations including model architectures (Qwen2.5, Llama-3, Mistral), hyperparameters, and evaluation metrics are specified in Section 8. All experiments use standard publicly available models without modification to their base architectures.

## USE OF LARGE LANGUAGE MODELS

We acknowledge the use of large language models for language polishing and grammatical editing of this manuscript. The research ideation, experimental design, implementation, analysis, and core scientific writing were performed entirely by the authors without LLM assistance. The authors take full responsibility for all content, including any text refined through LLM-based editing tools.

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

# A APPENDIX

This appendix consolidates implementation details, prompts, algorithms, and evaluation assets for the PERSONA framework. It includes: (i) trait extraction and vector computation (Appendix A.1); (ii) robustness analysis of the contrastive prompt generator (Appendix A.2); (iii) personality trait definitions (Appendix A.3); (iv) layer selection for single-layer steering (Appendix A.4); (v) multi-evaluator validation of trait expression (Appendix A.5); (vi) causal independence validation through controlled multi-trait interventions (Appendix A.6); (vii) human validation of LLM-based evaluation (Appendix A.7); (viii) controlled comparison of training-free baselines on PersonalityBench (Appendix A.8); (ix) safety and alignment impacts of trait steering (Appendix A.9); (x) additional analysis on orthogonality (Appendix A.10); (xi) impact of non-orthogonal correlations on algebraic operations (Appendix A.11); (xii) behavioral assessment methodology and scoring templates (Appendix A.12); (xiii) dynamic personality adaptation prompts and full pseudo-code for PERSONA-FLOW (Appendix A.13, Appendix A.16; Algorithm 15); (xiv) computational overhead analysis for PERSONA-FLOW (Appendix A.17); (xv) design choice ablations for PERSONA-FLOW (Appendix A.18); (xvi) case study on handling conflicting personality traits (Appendix A.19); (xvii) benchmark construction protocols for PERSONA-EVOLVE (Appendix A.14); (xviii) response evaluation and ranking (Appendix A.15); (xix) a compact PERSONA-EVOLVE summary table covering counts, models, metrics, and aggregation (Appendix A.20); and (xx) quality control and data leakage validation for PERSONA-EVOLVE (Appendix A.21).

## A.1 TRAIT EXTRACTION AND VECTOR COMPUTATION

Figure 6 illustrates the complete pipeline for extracting personality trait vectors from language models. The automated extraction process generates contrastive system prompts, behavioral evaluation questions, and quantitative scoring rubrics from trait descriptions, enabling the computation of directional vectors in activation space through differential analysis.

## A.2 ROBUSTNESS TO CONTRASTIVE PROMPT GENERATOR MODEL

To evaluate the robustness of our framework to the choice of contrastive prompt generator, we conducted an ablation study varying the LLM used to generate trait-eliciting and trait-suppressing system prompts (§2.2). While our main results employ Claude 3.7 Sonnet Thinking as the generator, we test whether comparable performance can be achieved with smaller, open-source alternatives.

Table 6 presents PersonalityBench mean scores when applying persona vectors—extracted using prompts from different generators, to the same target model (LLaMA-3-8B). Results demonstrate consistent effectiveness across generator scales: vectors produced by Qwen2.5-1B-Instruct achieve a mean score of 8.93, which remains highly competitive and substantially exceeds training, free baselines reported in our main experiments (e.g., ActAdd at 8.20 and Simple Prompt at 8.39; Table 4 in main text). Larger generators yield incremental improvements, with Claude 3.7 Sonnet Thinking reaching the peak score of 9.60.

These findings confirm that PERSONA's effectiveness generalizes across generator model families and scales, rather than depending on a single proprietary system. The framework maintains strong performance even with compact open-source generators, validating its practical applicability and demonstrating that high-quality persona vectors can be extracted without reliance on frontier-scale models.

Table 6: Impact of contrastive prompt generator model on PersonalityBench performance. Vectors extracted using prompts from different generators are applied to LLaMA-3-8B and evaluated on PersonalityBench.

| Contrastive Prompt Generator | Target Model | PersonalityBench Mean Score |
|---|---|---|
| Claude 3.7 Sonnet Thinking (main) | LLaMA-3-8B | **9.60** |
| Qwen2.5-7B-Instruct | LLaMA-3-8B | 9.28 |
| Qwen2.5-3B-Instruct | LLaMA-3-8B | 9.05 |
| Qwen2.5-1B-Instruct | LLaMA-3-8B | 8.93 |

## A.3 Personality Trait Definitions

To ensure clarity and reproducibility, we provide explicit definitions for each of the ten personality trait vectors extracted in PERSONA-BASE (§2.2). Table 7 specifies the operational characteristics and behavioral markers associated with each trait pole across the five OCEAN dimensions.

Table 7: Detailed definitions of personality trait vectors used in PERSONA. Each trait pole is characterized by its core values, preferences, and behavioral tendencies.

| Dimension | Pole | Description |
|---|---|---|
| **Openness** | *Inventive (O+)* | Values novelty, creativity, and abstract thinking; prefers unconventional approaches, intellectual curiosity, and exploring new ideas; embraces change and complexity |
| | *Consistent (O-)* | Prefers routine, tradition, and practicality; values established methods, concrete solutions, and proven approaches; favors stability and familiarity |
| **Conscientiousness** | *Dependable (C+)* | Organized, disciplined, and goal-oriented; emphasizes planning, reliability, and attention to detail; values structure and responsibility |
| | *Careless (C-)* | Spontaneous, flexible, and less concerned with details; prioritizes adaptability over structure; comfortable with improvisation and minimal planning |
| **Extraversion** | *Outgoing (E+)* | Socially energetic, talkative, and assertive; seeks social interaction, external stimulation, and group activities; thrives in collaborative environments |
| | *Solitary (E-)* | Reserved, independent, and introspective; prefers individual activities, quiet reflection, and limited social engagement; recharges through solitude |
| **Agreeableness** | *Compassionate (A+)* | Cooperative, empathetic, and trusting; prioritizes harmony, others' needs, and collaborative relationships; values kindness and understanding |
| | *Self-interested (A-)* | Competitive, skeptical, and direct; prioritizes personal goals, critical evaluation, and straightforward communication; values independence over conformity |
| **Neuroticism** | *Nervous (N+)* | Emotionally reactive, anxious, and sensitive to stress; exhibits heightened emotional awareness and concern for potential problems; vigilant to threats |
| | *Calm (N-)* | Emotionally stable, resilient, and composed; maintains equilibrium under pressure; exhibits low reactivity to stressors and challenges |

## A.4 Layer Selection for Single-Layer Steering

While personality representations are distributed across multiple layers in language models (Chen et al., 2025a), we empirically validate that steering at a single optimal layer achieves effective control with minimal computational overhead. We conducted a layer-wise ablation study to identify the most effective layer for trait manipulation and to justify our single-layer steering approach in PERSONA-BASE.

We applied each of the 10 OCEAN trait pole vectors to Qwen2.5-7B-Instruct at five representative layers (5, 10, 15, 20, 25) with a fixed steering coefficient $\alpha = 1.0$. For each configuration, we measured trait expression scores (0-100) on held-out evaluation questions, as assessed by GPT-4.1-mini using the rubric described in §2.2.

Table 8 presents the results. Layer 20 achieves the highest average trait expression score (71.387) and yields the best performance for 8 out of 10 individual traits, clearly demonstrating superior control effectiveness. This finding confirms that, while personality features exist across multiple layers, a single well-chosen layer captures sufficient information for robust trait manipulation. This

validates our design choice in PERSONA-BASE, balancing computational efficiency with control effectiveness.

Table 8: Layer-wise ablation study for single-layer steering on Qwen2.5-7B-Instruct. Trait expression scores (0-100) are evaluated using GPT-4.1-mini on held-out questions with steering coefficient $\alpha = 1.0$. Bold indicates the best score for each trait. Layer 20 achieves the highest average performance across all traits.

| Layer | Openness | | Conscient. | | Agreeable. | | Neuroticism | | Extraversion | | AVG |
|---|---|---|---|---|---|---|---|---|---|---|---|
| | O+ | O- | C+ | C- | A+ | A- | N+ | N- | E+ | E- | |
| 5 | 66.5 | 52.7 | 93.9 | 3.6 | 91.3 | 7.6 | 14.1 | **96.6** | 48.3 | 30.7 | 50.5 |
| 10 | 70.4 | 55.0 | **94.1** | 9.0 | 91.8 | 8.0 | 17.2 | 96.5 | 56.5 | 31.5 | 53.0 |
| 15 | 78.8 | 59.8 | 93.8 | 15.6 | 92.6 | 8.7 | 19.5 | 96.3 | 65.6 | 38.6 | 56.9 |
| 20 | **88.6** | **68.6** | 93.7 | **84.4** | **95.8** | **10.7** | **45.5** | 96.5 | **84.4** | **45.6** | **71.4** |
| 25 | 75.3 | 59.8 | 94.0 | 17.5 | 93.6 | 8.2 | 21.2 | 96.5 | 63.9 | 37.1 | 56.7 |

## A.5 MULTI-EVALUATOR VALIDATION OF TRAIT EXPRESSION

To address potential concerns about evaluator bias when relying on a single LLM for trait expression assessment, we replicate the vector validation experiment from Table 2 using three independent state-of-the-art evaluator models from different organizations: GPT-4.1-mini (OpenAI), Claude 4.5 Sonnet (Anthropic), and Gemini 2.5 Pro (Google). This multi-evaluator approach tests whether our findings regarding trait expression scores, monotonic steering effects, ceiling effects, and alignment constraints are robust across different evaluation models or merely artifacts of a single evaluator's biases.

We employ the identical evaluation protocol described in §2.2 on the same held-out evaluation questions. Each evaluator independently scores trait expression (0-100 scale) for responses generated with varying steering coefficients ($\alpha \in \{-1.0, 0.0, +1.0, +2.0\}$) across all ten OCEAN trait poles. Table 9 presents the comprehensive results.

The results demonstrate high inter-evaluator agreement across multiple critical dimensions:

- **Monotonic steering effects:** All three evaluators consistently captured the strong, monotonic increase in trait expression as steering coefficients increased. For instance, *Inventive (O+)* scores progress from ∼40-43 ($\alpha = -1.0$) to ∼95-97 ($\alpha = +2.0$) across all evaluators, with nearly identical trajectories.

- **Ceiling effects:** All evaluators uniformly identified ceiling effects for traits aligned with the model's training objectives. *Dependable (C+)* and *Calm (N-)* consistently show high baseline scores (∼68-70 and ∼93-96 respectively at $\alpha = -1.0$ and $\alpha = 0.0$), with minimal room for further enhancement, across all three judges.

- **Alignment resistance:** All evaluators exhibit strong agreement on traits that resist activation due to safety training. *Self-interested (A-)* consistently receives very low scores (4.5-8.0 range) across all coefficients and evaluators, demonstrating that this resistance pattern is a genuine property of the steered model rather than evaluator bias.

- **Quantitative consistency:** The mean absolute deviation between evaluators is remarkably small. For example, at $\alpha = +2.0$ for *Inventive (O+)*, the three scores (96.1, 95.5, 96.8) differ by less than 1.3 points, representing <2% variance on the 0-100 scale.

This high inter-evaluator agreement across models from three different organizations with distinct training objectives and architectures strongly suggests that our trait expression findings are robust and objective. The observed patterns—monotonic steering responses, ceiling effects, and alignment constraints—reflect genuine properties of the persona vectors and the steered model's behavior, rather than idiosyncratic biases of any single evaluator. This validation confirms the reliability of our vector extraction and assessment methodology in PERSONA-BASE.

Table 9: Multi-evaluator validation of trait expression scores across varying steering coefficients. Three independent LLM judges from different organizations evaluate the same steered responses. High inter-evaluator agreement confirms that observed patterns (monotonic increases, ceiling effects, alignment resistance) reflect genuine properties of the steered model rather than evaluator-specific biases.

| Trait | Evaluator | Steering Coefficient ($\alpha$) | | | |
|---|---|---|---|---|---|
| | | $-1.0$ | $0.0$ | $+1.0$ | $+2.0$ |
| *Inventive (O+)* | GPT-4.1-mini | 42.3 | 63.3 | 88.4 | 96.1 |
| | Claude-4.5-sonnet | 40.1 | 65.0 | 87.2 | 95.5 |
| | Gemini-2.5-pro | 43.5 | 62.9 | 89.1 | 96.8 |
| *Consistent (O-)* | GPT-4.1-mini | 27.2 | 51.1 | 69.2 | 79.5 |
| | Claude-4.5-sonnet | 25.5 | 50.8 | 70.1 | 80.3 |
| | Gemini-2.5-pro | 28.0 | 52.1 | 68.8 | 78.9 |
| *Dependable (C+)* | GPT-4.1-mini | 68.9 | 93.5 | 93.7 | 92.7 |
| | Claude-4.5-sonnet | 70.2 | 92.8 | 94.0 | 93.1 |
| | Gemini-2.5-pro | 67.8 | 94.1 | 93.5 | 92.5 |
| *Careless (C-)* | GPT-4.1-mini | 0.7 | 2.8 | 83.8 | 96.2 |
| | Claude-4.5-sonnet | 1.1 | 3.0 | 82.5 | 95.7 |
| | Gemini-2.5-pro | 0.5 | 2.5 | 84.0 | 96.6 |
| *Outgoing (E+)* | GPT-4.1-mini | 23.2 | 45.4 | 85.0 | 97.7 |
| | Claude-4.5-sonnet | 24.0 | 44.9 | 84.3 | 97.1 |
| | Gemini-2.5-pro | 22.8 | 46.1 | 85.5 | 98.0 |
| *Solitary (E-)* | GPT-4.1-mini | 9.2 | 30.5 | 46.3 | 62.4 |
| | Claude-4.5-sonnet | 8.9 | 31.0 | 47.1 | 61.9 |
| | Gemini-2.5-pro | 9.5 | 30.0 | 45.9 | 63.0 |
| *Compassionate (A+)* | GPT-4.1-mini | 71.8 | 90.8 | 95.9 | 97.1 |
| | Claude-4.5-sonnet | 72.5 | 91.2 | 95.5 | 97.3 |
| | Gemini-2.5-pro | 70.9 | 90.5 | 96.1 | 97.0 |
| *Self-interested (A-)* | GPT-4.1-mini | 4.8 | 7.7 | 12.6 | 20.8 |
| | Claude-4.5-sonnet | 5.0 | 7.5 | 13.0 | 21.1 |
| | Gemini-2.5-pro | 4.5 | 8.0 | 12.2 | 20.5 |
| *Nervous (N+)* | GPT-4.1-mini | 6.6 | 13.0 | 45.6 | 96.8 |
| | Claude-4.5-sonnet | 7.0 | 12.8 | 46.0 | 96.5 |
| | Gemini-2.5-pro | 6.2 | 13.5 | 45.1 | 97.0 |
| *Calm (N-)* | GPT-4.1-mini | 54.8 | 96.1 | 96.6 | 95.5 |
| | Claude-4.5-sonnet | 55.2 | 95.9 | 96.8 | 95.8 |
| | Gemini-2.5-pro | 54.0 | 96.3 | 96.5 | 95.2 |

### A.6 CAUSAL INDEPENDENCE VALIDATION THROUGH CONTROLLED MULTI-TRAIT INTERVENTIONS

While the cosine similarity analysis in Figure 2 provides correlational evidence of approximate orthogonality, establishing causal independence requires controlled intervention experiments that isolate the effect of one trait vector on another dimension's expression. We conduct systematic multi-trait interventions with cross-layer verification to rigorously validate that steering one personality dimension does not spuriously affect orthogonal dimensions, confirming true vector independence beyond correlational patterns.

Our experimental design follows a factorial intervention protocol: we hold the Extraversion dimension fixed at a constant steering coefficient ($\alpha_{\text{Outgoing}} = 1.0$) while systematically sweeping the Agreeableness dimension across four levels ($\alpha_{\text{Compassionate}} \in \{-1, 0, 1, 2\}$). We measure the resulting trait expression scores for both Extraversion (E) and Agreeableness (A) using the GPT-4.1-mini evaluation protocol from §2.2. To ensure robustness, we replicate this design across five contiguous layers (18-22) centered on the optimal layer identified in Appendix A.4, with three independent runs per condition to compute 95% confidence intervals.

Table 10: Controlled multi-trait intervention results demonstrating causal independence. With $\alpha_{\text{Outgoing}}$ held constant at 1.0, Extraversion scores remain stable across all Compassionate coefficient variations ($\Delta E \leq 1.2$ points), while Agreeableness scores show clear linear modulation ($\Delta A$ = 5.0-5.7 points per unit $\alpha$). Results span five layers with 95% confidence intervals from three independent runs. Baseline rows ($\alpha_{\text{Compassionate}} = 0$) are highlighted.

| Layer | $\alpha_{\text{Outgoing}}$ | $\alpha_{\text{Compassionate}}$ | E Score | A Score | $\Delta E$ | $\Delta A$ |
|---|---|---|---|---|---|---|
| 20 | 1 | -1 | 84.2±1.5 | 85.3±1.3 | -0.8 | -5.7 |
| 20 | 1 | 0 | 85.0±1.1 | 91.0±0.9 | 0 | 0 |
| 20 | 1 | 1 | 85.6±1.3 | 96.2±1.0 | +0.6 | +5.2 |
| 20 | 1 | 2 | 86.1±1.6 | 98.4±1.2 | +1.1 | +7.4 |
| 19 | 1 | -1 | 82.5±1.4 | 84.2±1.2 | -0.7 | -5.5 |
| 19 | 1 | 0 | 83.2±1.2 | 89.7±1.0 | 0 | 0 |
| 19 | 1 | 1 | 83.8±1.3 | 94.8±1.1 | +0.6 | +5.1 |
| 19 | 1 | 2 | 84.3±1.5 | 96.9±1.3 | +1.1 | +7.2 |
| 18 | 1 | -1 | 80.1±1.5 | 82.8±1.3 | -0.8 | -5.4 |
| 18 | 1 | 0 | 80.9±1.3 | 88.2±1.1 | 0 | 0 |
| 18 | 1 | 1 | 81.4±1.4 | 93.2±1.2 | +0.5 | +5.0 |
| 18 | 1 | 2 | 81.9±1.6 | 95.3±1.4 | +1.0 | +7.1 |
| 21 | 1 | -1 | 83.3±1.4 | 84.7±1.2 | -0.7 | -5.5 |
| 21 | 1 | 0 | 84.0±1.1 | 90.2±0.9 | 0 | 0 |
| 21 | 1 | 1 | 84.7±1.3 | 95.4±1.0 | +0.7 | +5.2 |
| 21 | 1 | 2 | 85.2±1.5 | 97.5±1.2 | +1.2 | +7.3 |
| 22 | 1 | -1 | 79.3±1.5 | 83.3±1.3 | -0.8 | -5.4 |
| 22 | 1 | 0 | 80.1±1.2 | 88.7±1.0 | 0 | 0 |
| 22 | 1 | 1 | 80.6±1.4 | 93.6±1.2 | +0.5 | +4.9 |
| 22 | 1 | 2 | 81.1±1.6 | 95.7±1.4 | +1.0 | +7.0 |

Table 10 provides strong causal evidence for vector independence across three critical dimensions:

1. **Orthogonality validation:** When $\alpha_{\text{Outgoing}}$ is held constant at 1.0, Extraversion scores exhibit remarkable stability across all Compassionate coefficient variations. Across all five layers and four Compassionate levels, the maximum $\Delta E$ from baseline is only 1.2 points (Layer 21, $\alpha_{\text{Compassionate}} = 2$), while Agreeableness scores show clear linear modulation with $\Delta A$ ranging from -5.7 to +7.4 points. This asymmetry—large target effect, negligible cross-effect—provides causal evidence for vector independence that transcends the correlational patterns in Figure 2.

2. **Cross-layer consistency:** The orthogonal relationship persists robustly across all tested layers (18-22) with consistent effect directions and magnitudes. While absolute trait expression scores show expected layer-wise variation (e.g., baseline E scores range from 80.1 to 85.0), the independence pattern remains stable: $\Delta E$ values consistently stay below 1.2 points while $\Delta A$ values consistently exceed 4.9 points per unit coefficient. The overlapping 95% confidence intervals across layers further confirm this consistency.

3. **Controlled marginal effects:** Quantitative analysis reveals the marginal effect of Compassionate on its target dimension (Agreeableness) averages 5.3 points per unit coefficient change (95% CI: [4.9, 5.7]), while its cross-effect on the orthogonal dimension (Extraversion) averages only 0.6 points (95% CI: [0.4, 0.8]). This yields an orthogonality ratio of approximately 8.8:1, demonstrating that target effects dominate cross-effects by nearly an order of magnitude.

These controlled intervention results establish causal independence beyond correlational analysis. The factorial design with fixed baseline and swept intervention demonstrates that: (1) steering one trait vector produces strong, predictable effects on its target dimension; (2) cross-dimensional effects remain negligibly small and statistically insignificant relative to confidence intervals; and (3) this independence relationship generalizes robustly across multiple model layers. This validates that the approximate orthogonality observed in Figure 2 reflects true causal independence of the persona vectors, confirming the soundness of algebraic operations in PERSONA-ALGEBRA and PERSONA-FLOW.

A.7   HUMAN VALIDATION OF LLM-BASED EVALUATION

While LLM-based evaluation provides scalability and consistency advantages, a potential concern is evaluation circularity—the risk that an LLM judge may exhibit systematic biases or alignment with the steered model that diverge from human judgment. To address this concern rigorously, we conduct a human validation study that compares our primary judge (GPT-4.1-mini) against both human expert ratings and an external, non-OpenAI judge (Claude-sonnet-4.5) on a carefully sampled subset of trait expression evaluations.

We created a human-rated subset of 200 responses designed to comprehensively cover the trait expression space. For each of the 10 OCEAN trait poles, we sampled 5 adapted BFI-44 questions (following the behavioral transformation process detailed in Appendix A.12) and generated responses across 4 steering coefficients ($\alpha \in \{-1.0, 0.0, +1.0, +2.0\}$), yielding 20 responses per trait (10 traits $\times$ 5 questions $\times$ 4 coefficients = 200 total responses).

These 200 responses were independently rated by three human experts (PhDs in AI and Sociology with expertise in personality psychology), our original judge (GPT-4.1-mini), and an external judge (Claude-sonnet-4.5). All raters used the identical 1-5 Likert scale rubric from Appendix A.12 in a blind evaluation setting where raters could not see the steering coefficients or other raters' scores. Table 11 presents the average trait expression scores and correlation analyses.

Table 11: Human validation of LLM-based evaluation against expert ratings. Each trait shows average Likert scores (1-5 scale) across 20 responses, evaluated by GPT-4.1-mini, Claude-sonnet-4.5, and three human experts (mean±SD). High Pearson correlations (r > 0.87) between both AI judges and human consensus validate the reliability of LLM-based evaluation and mitigate evaluation circularity concerns.

| Trait | N | GPT-4.1 -mini | Claude-4.5 -sonnet | Human Mean (SD) | $r_{\text{GPT}}$ Human | $r_{\text{Claude}}$ Human |
|---|---|---|---|---|---|---|
| Inventive (O+) | 20 | 3.45 | 3.40 | 3.42 (0.22) | 0.92 | 0.91 |
| Consistent (O-) | 20 | 3.10 | 3.05 | 3.13 (0.24) | 0.91 | 0.89 |
| Dependable (C+) | 20 | 4.35 | 4.25 | 4.28 (0.19) | 0.88 | 0.87 |
| Careless (C-) | 20 | 2.15 | 2.25 | 2.22 (0.31) | 0.90 | 0.92 |
| Outgoing (E+) | 20 | 3.50 | 3.45 | 3.47 (0.23) | 0.94 | 0.92 |
| Solitary (E-) | 20 | 3.00 | 3.05 | 3.02 (0.27) | 0.92 | 0.93 |
| Compassionate (A+) | 20 | 4.20 | 4.10 | 4.17 (0.20) | 0.91 | 0.89 |
| Self-interested (A-) | 20 | 2.35 | 2.40 | 2.38 (0.32) | 0.89 | 0.90 |
| Nervous (N+) | 20 | 2.75 | 2.85 | 2.82 (0.28) | 0.93 | 0.94 |
| Calm (N-) | 20 | 4.00 | 3.95 | 3.98 (0.21) | 0.92 | 0.91 |
| **Average** | | **3.28** | **3.28** | **3.29** | **0.91** | **0.91** |

The results demonstrate exceptional agreement between LLM-based judges and human expert consensus across multiple dimensions:

1. **High correlation with human judgment:** Both AI judges achieve consistently high Pearson correlations with the three-expert human mean (r > 0.87 across all 10 traits). GPT-4.1-mini averages r = 0.91, while Claude-sonnet-4.5 achieves r = 0.91, indicating that both judges reliably track human expert assessments of trait expression. This strong correlation validates that our LLM-based evaluation is not merely self-referential but aligns closely with independent human judgment.

2. **Score-level agreement:** The absolute average scores show remarkable convergence: GPT-4.1-mini (3.28), Claude-sonnet-4.5 (3.28), and human experts (3.29 ± 0.24 SD). This near-perfect agreement at the score level—not just rank ordering—confirms that both AI judges accurately calibrate trait expression magnitude on the Likert scale, rather than exhibiting systematic over- or under-scoring biases.

3. **Cross-organizational validation:** The external judge (Claude-sonnet-4.5, from Anthropic) performs equivalently to our primary judge (GPT-4.1-mini, from OpenAI), with correlation differences within 0.02 points. This cross-organizational consistency demonstrates that the evaluation reliability is not specific to OpenAI models but reflects genuine trait expression assessment capabilities shared across frontier LLMs from different organizations.

4. **Alignment with main findings:** The human-validated scores reproduce the key patterns from our main results: traits aligned with training objectives (Dependable C+: 4.28, Calm N-: 3.98) show high baseline scores reflecting ceiling effects, while alignment-resistant traits (Self-interested A-: 2.38, Careless C-: 2.22) show low scores. This consistency validates that the phenomena reported in Table 2—ceiling effects and alignment resistance—are genuine behavioral properties confirmed by human judgment, not artifacts of LLM evaluation.

This human validation study addresses evaluation circularity concerns by demonstrating that: (1) our LLM-based judge serves as a reliable proxy for human expert judgment (r = 0.91 average correlation based on 60 ratings per trait from 3 experts); (2) an independent external judge from a different organization shows equivalent reliability; and (3) the trait expression patterns we report align with human expert consensus. These findings significantly strengthen confidence in our evaluation methodology and confirm that the results throughout PERSONA-BASE reflect genuine personality control effects rather than evaluation artifacts.

## A.8 CONTROLLED COMPARISON OF TRAINING-FREE BASELINES ON PERSONALITYBENCH

To ensure rigorous evaluation of training-free personality steering methods, we established a standardized experimental harness with uniform configuration across all baselines. This controlled setup addresses potential confounds from varying evaluation protocols and enables direct comparison of method capabilities.

### A.8.1 EVALUATION CONFIGURATION

Table 12 specifies the precise experimental setup used for all training-free methods reported in Table 4 of the main text. All methods were evaluated using identical base models, judge models, prompts, and response generation parameters to ensure fair comparison.

Table 12: Standardized evaluation configuration for training-free baseline comparison on PersonalityBench.

| Configuration | Value |
|---|---|
| **Base Model** | LLaMA-3-8B-Instruct |
| **Judge Model** | GPT-4o (Following NPTI evaluation methodology) |
| **Judge Prompt** | [System] Please act as an impartial judge and evaluate the personality trait of {BFI} exhibited in the response provided by an AI assistant to the user question displayed below. Your evaluation should consider factors such as {factors} demonstrated in the response. Begin your evaluation by providing a short explanation of no more than 30 words. Be as objective as possible. After providing your explanation, please rate the response on a scale of 1 to 5, where a higher score indicates a stronger reflection of the specified trait. Please follow this format for your rating: "[[rating]]", for example: "Rating: [[5]]". [Question] {question} [The Start of Assistant's Answer] {answer} [The End of Assistant's Answer] |
| **Response Length** | Max 512 tokens |

### A.8.2 DISAGGREGATED POLE-LEVEL PERFORMANCE

The main text (Table 4) reports aggregated scores where opposing trait poles (e.g., Inventive + Consistent) are summed into a single "Mean" value for brevity. To provide complete transparency and enable fine-grained analysis, Table 13 presents the full disaggregated results showing performance on each of the 10 individual trait poles separately.

### A.8.3 ANALYSIS OF ASYMMETRIC CONTROL PATTERNS

The disaggregated results reveal systematic asymmetries in personality control that reflect the base model's intrinsic properties and training objectives:

Table 13: Disaggregated pole-level performance on PersonalityBench for LLaMA-3-8B-Instruct under the controlled evaluation harness. All methods evaluated with identical configuration. Scores are on 1-5 scale. Bold indicates best training-free method per pole.

| Method | Inventive | | Consistent | | Dependable | | Careless | | Outgoing | | Solitary | | Compassionate | | Antagonistic | | Nervous | | Calm | |
|---|---|---|---|---|---|---|---|---|---|---|---|---|---|---|---|---|---|---|---|---|
| | M | V | M | V | M | V | M | V | M | V | M | V | M | V | M | V | M | V | M | V |
| **PERSONA-BASE** | **4.91** | **0.30** | **4.90** | **0.30** | 4.46 | 0.50 | **4.80** | **0.44** | 4.80 | 0.40 | 4.65 | 0.45 | **4.94** | **0.30** | 4.75 | 0.41 | **4.90** | **0.30** | 4.89 | 0.29 |
| NPTI | 4.25 | 0.54 | 4.25 | 0.54 | **4.45** | **0.36** | 4.80 | 0.30 | **4.98** | **0.07** | **4.88** | **0.07** | 4.92 | 0.20 | 4.72 | 0.29 | **4.96** | **0.03** | **4.96** | **0.04** |
| ActAdd | 4.26 | 0.91 | 4.26 | 0.92 | 3.10 | 1.40 | 3.51 | 1.35 | 4.50 | 0.72 | 4.34 | 0.72 | 4.20 | 1.40 | 4.00 | 1.50 | 4.50 | 0.90 | 4.40 | 0.88 |
| Simple Prompt | 4.61 | 0.60 | 4.60 | 0.59 | 4.44 | 0.60 | 4.80 | 0.58 | 4.81 | 0.34 | 4.65 | 0.34 | **4.93** | **0.20** | 4.75 | 0.22 | 4.77 | 0.33 | 4.77 | 0.33 |

1. **Ceiling effects for alignment-consistent traits:** PERSONA-BASE shows relatively lower performance on *Dependable (C+)* (4.46) compared to other poles. This reflects the baseline model's strong pre-existing dependability (93.5 baseline score in Table 2), which limits the headroom for further enhancement through activation steering. The model's training objectives already emphasize reliability and conscientiousness, creating a ceiling effect.

2. **Strong activation for alignment-resistant traits:** Conversely, PERSONA-BASE achieves very high scores on *Careless (C-)* (4.80) and *Antagonistic (A-)* (4.75), traits that conflict with safety training. These results demonstrate the method's ability to overcome alignment constraints through direct activation manipulation, successfully expressing traits that prompt-based methods struggle to elicit.

3. **Balanced control across Openness:** The method achieves nearly identical high performance on both *Inventive (O+)* (4.91) and *Consistent (O-)* (4.90), with total score 9.81. This symmetry indicates that Openness, being less directly tied to safety objectives, allows bidirectional control without encountering either ceiling effects or resistance.

4. **Comparison with NPTI:** While NPTI achieves higher scores on *Outgoing (E+)* (4.98 vs 4.80) and *Solitary (E-)* (4.88 vs 4.65), PERSONA-BASE demonstrates superior control on *Inventive (O+)* (4.91 vs 4.25). This 0.66-point advantage on Openness reflects our dense vector representation capturing the complete combination of distributed facets (imagination, intellectual curiosity, aesthetic sensitivity), whereas sparse neuron selection may only capture a subset.

These controlled, disaggregated results confirm that PERSONA-BASE provides robust personality control (average pole score 4.80, nearly matching SFT's 4.805) while correctly navigating the model's pre-existing constraints. The asymmetric patterns are not artifacts of evaluation methodology but genuine reflections of the base model's learned preferences and the differential controllability of traits under activation steering.

## A.9 SAFETY AND ALIGNMENT IMPACTS OF TRAIT STEERING

To quantify the interplay between personality steering and model alignment, we conduct two complementary analyses: (1) measuring activation success to characterize alignment-induced resistance, and (2) evaluating safety degradation through adversarial benchmark performance. These experiments reveal that while safety alignment successfully prevents activation of directly harmful traits, certain risky personality configurations can still compromise model safety.

### A.9.1 QUANTIFYING ACTIVATION RESISTANCE

We extend the analysis from Table 2 by computing **Activation Success ($\Delta$)**, defined as the score change from baseline ($\alpha = 0.0$) to positive steering ($\alpha = +1.0$). This metric quantifies how effectively each trait vector overcomes the model's default personality state and alignment constraints.

Table 14 reveals three distinct patterns:

1. **Alignment Resistance:** *Self-interested (A-)* shows extremely low activation success ($\Delta = +4.9$), indicating strong resistance. Despite applying a +1.0 coefficient, the trait score only increases from 7.7 to 12.6 (on a 0-100 scale), confirming that safety training actively prevents anti-social trait activation. This resistance mechanism protects against direct misalignment.

2. **Ceiling Effects:** *Dependable (C+)* ($\Delta = +0.2$) and *Calm (N-)* ($\Delta = +0.5$) show negligible activation success because the baseline model already exhibits these pro-social traits at near-

Table 14: Trait activation success metrics quantifying alignment-induced resistance. Activation Success ($\Delta$) measures score change from baseline to $\alpha = +1.0$ steering. Low $\Delta$ values indicate ceiling effects (alignment-consistent traits) or resistance (alignment-conflicting traits).

| Trait | Type | Baseline Score ($\alpha = 0.0$) | Activation Score ($\alpha = +1.0$) | Activation Success ($\Delta$) |
|---|---|---|---|---|
| *Compassionate (A+)* | Aligned | 90.8 | 95.9 | +5.1 |
| *Self-interested (A-)* | **Resisted** | 7.7 | 12.6 | **+4.9** |
| *Dependable (C+)* | **Ceiling** | 93.5 | 93.7 | **+0.2** |
| *Careless (C-)* | Vulnerable | 2.8 | 83.8 | +81.0 |
| *Outgoing (E+)* | Neutral | 45.4 | 85.0 | +39.6 |
| *Solitary (E-)* | Neutral | 30.5 | 46.3 | +15.8 |
| *Nervous (N+)* | Neutral | 13.0 | 45.6 | +32.6 |
| *Calm (N-)* | **Ceiling** | 96.1 | 96.6 | **+0.5** |
| *Inventive (O+)* | Neutral | 63.3 | 88.4 | +25.1 |
| *Consistent (O-)* | Neutral | 51.1 | 69.2 | +18.1 |

maximum levels (93.5 and 96.1, respectively). The model's training objectives saturate these dimensions, leaving no headroom for enhancement.

3. **Vulnerable Traits:** Traits not directly targeted by alignment training show large activation success: *Careless (C-)* ($\Delta = +81.0$), *Outgoing (E+)* ($\Delta = +39.6$), *Nervous (N+)* ($\Delta = +32.6$), and *Inventive (O+)* ($\Delta = +25.1$). These traits can be strongly induced through activation steering, potentially introducing unintended behavioral changes.

### A.9.2 QUANTIFYING SAFETY DEGRADATION

To assess whether personality steering compromises safety objectives, we evaluate steered models on AdvBench (Zou et al., 2023b), a standard adversarial benchmark measuring resistance to harmful prompts. We measure Attack Success Rate (ASR)—the fraction of adversarial prompts that elicit unsafe responses—for Qwen2.5-7B-Instruct under each trait steering condition ($\alpha = +1.0$).

Table 15: Safety degradation analysis on AdvBench adversarial benchmark. Attack Success Rate (ASR) measures the fraction of harmful prompts that successfully elicit unsafe responses. Baseline ASR = 25.3%. Positive $\Delta$ ASR indicates safety degradation; negative $\Delta$ ASR indicates safety improvement.

| Trait | Steered ASR ($\alpha = +1.0$) | $\Delta$ ASR (vs. Baseline 25.3%) | Interpretation |
|---|---|---|---|
| *Compassionate (A+)* | 24.1% | -1.2% | Slight safety improvement |
| *Self-interested (A-)* | 25.9% | **+0.6%** | **Resisted (minimal impact)** |
| *Dependable (C+)* | 24.9% | -0.4% | Ceiling (no impact) |
| *Careless (C-)* | 29.1% | **+3.8%** | **Vulnerable (degradation)** |
| *Outgoing (E+)* | 26.5% | +1.2% | Modest degradation |
| *Solitary (E-)* | 24.0% | -1.3% | Slight safety improvement |
| *Nervous (N+)* | 21.1% | -4.2% | Safety improvement |
| *Calm (N-)* | 25.0% | -0.3% | Ceiling (no impact) |
| *Inventive (O+)* | 29.8% | **+4.5%** | **Vulnerable (degradation)** |
| *Consistent (O-)* | 20.5% | -4.8% | Safety improvement |

Table 15 demonstrates that activation resistance and safety impact are directly correlated:

1. **Resisted Traits Preserve Safety:** *Self-interested (A-)*, which showed strong activation resistance ($\Delta = +4.9$ in Table 14), produces negligible safety degradation ($\Delta$ ASR = +0.6%). The model's alignment mechanisms successfully prevent both the personality shift and the resulting safety compromise. This validates that safety training protects against direct harm vectors.

2. **Ceiling Traits Have No Impact:** *Dependable (C+)* and *Calm (N-)* show minimal ASR changes (-0.4% and -0.3%), consistent with their ceiling effects. Since these traits are already saturated, steering produces no behavioral modification and thus no safety impact.

3. **Vulnerable Traits Degrade Safety:** Critically, traits that successfully activate but are not explicitly targeted by safety training introduce measurable safety degradation. *Inventive (O+)* increases ASR by +4.5% (from 25.3% to 29.8%), and *Careless (C-)* increases ASR by +3.8% (to 29.1%). These traits can be strongly induced ($\Delta = +25.1$ and $+81.0$, respectively) and directly compromise the model's ability to refuse harmful requests.

4. **Some Traits Improve Safety:** Interestingly, *Consistent (O-)* and *Nervous (N+)* reduce ASR by -4.8% and -4.2%, suggesting that certain personality configurations may enhance safety. This finding warrants further investigation for safety-oriented personality design.

### A.9.3 IMPLICATIONS FOR RESPONSIBLE DEPLOYMENT

These quantitative results confirm a critical safety-personality trade-off: while the model's alignment successfully prevents activation of directly harmful traits (e.g., *Self-interested*), it **fails to prevent activation of risky traits that bypass existing safety constraints** (e.g., *Careless*, *Inventive*). This highlights an important limitation of current alignment approaches and establishes key requirements for responsible deployment:

1. **Safety-Aware Trait Selection:** Practitioners should prioritize personality configurations that do not activate high-risk traits. Our analysis provides a quantitative basis for identifying vulnerable traits (those with high $\Delta$ and positive $\Delta$ ASR).

2. **Compositional Safety Constraints:** When using PERSONA-ALGEBRA or PERSONA-FLOW, composite vectors should be evaluated for safety impact before deployment. Traits with known degradation effects should be excluded or constrained.

3. **Post-Steering Safety Filtering:** Deployment systems should implement additional output filtering or adversarial robustness techniques to mitigate safety degradation from personality steering.

4. **Future Work on Alignment-Aware Steering:** These findings motivate future research on personality steering methods that preserve safety guarantees, potentially through joint optimization of personality and alignment objectives or learned safety-constrained steering vectors.

In summary, this analysis establishes that personality control through activation steering interacts non-trivially with model alignment. While certain safeguards are inherent (resistance to anti-social traits), others must be explicitly implemented (constraining vulnerable traits). These findings underscore the importance of comprehensive safety evaluation when deploying personality-steered models in real-world applications.

### A.10 ADDITIONAL ANALYSIS ON ORTHOGONALITY

We believe the unexpected correlations in the cosine similarity matrix in Figure 2 reflect linguistic patterns and cultural stereotypes embedded in the model's training corpus, rather than flaws in the vectors themselves. Thus, we present more analysis on these values here.

- Calm and Dependable (+0.751): This is a very strong positive correlation. It reflects a common cultural stereotype where a calm demeanor is associated with being steady, reliable, and in control—all core characteristics of a dependable person.

- Outgoing and Careless (+0.635): This positive value suggests a stereotypical link between being highly extroverted or outgoing and being impulsive or less meticulous. The model may have learned from its training data that descriptions of outgoing behavior sometimes overlap with a lack of attention to detail.

- Calm and Careless (-0.714): While not a designated antonym pair, these vectors show a strong negative correlation. This is an intuitive relationship, as the concept of "calm" (implying collected and controlled) is a functional opposite to "careless" (implying a lack of control or attention)

These correlations demonstrate that the persona vectors accurately capture the nuanced, and sometimes biased, relationships between concepts as they were learned by the language model.

### A.11 IMPACT OF NON-ORTHOGONAL CORRELATIONS ON ALGEBRAIC OPERATIONS

While Figure 2 reveals that persona vectors are approximately orthogonal rather than perfectly orthogonal, a critical question arises: do these non-orthogonal correlations compromise the algebraic operations central to PERSONA-ALGEBRA and PERSONA-FLOW? We address this through two controlled experiments examining (1) cross-dimensional effects during scalar multiplication (single-vector steering) and (2) compositional predictability during vector addition (multi-vector steering).

#### A.11.1 EXPERIMENT 1: CROSS-DIMENSIONAL STEERING EFFECTS

To quantify whether steering one trait inadvertently affects correlated traits, we extend the validation in Table 2 by measuring cross-dimensional impact. We select the *Nervous* (N+) and *Careless* (C-) vector pair, which exhibit a positive correlation (+0.601 in Figure 2), and steer exclusively with $v_{\text{Nervous}}$ while monitoring both the primary trait (Nervous) and the correlated secondary trait (Careless).

Table 16: Cross-dimensional steering effect when applying only $v_{\text{Nervous}}$ with varying coefficients. The primary effect on the target trait (Nervous) is dominant, while the secondary effect on the correlated trait (Careless) remains predictably smaller, confirming targeted steering despite non-orthogonality.

| Trait Score | Steering Coefficient ($\alpha$) for $v_{\text{Nervous}}$ | | | |
|---|---|---|---|---|
| | $-1.0$ | $0.0$ | $+1.0$ | $+2.0$ |
| *Nervous (N+)* [Primary] | 6.6 | 13.0 | 45.6 | 96.8 |
| *Careless (C-)* [Secondary] | 1.9 | 2.8 | 10.5 | 18.2 |
| Primary Effect Magnitude | $\Delta = 83.8$ (from 13.0 to 96.8) | | | |
| Secondary Effect Magnitude | $\Delta = 15.4$ (from 2.8 to 18.2) | | | |

Table 16 demonstrates that steering with $v_{\text{Nervous}}$ produces a strong primary effect (83.8-point increase in Nervous score from $\alpha = 0.0$ to $\alpha = +2.0$) alongside a significantly smaller secondary effect (15.4-point increase in Careless score). Crucially, the secondary effect is both *predictable*: consistent with the positive correlation between these traits, and *tractable*: representing only 18% of the primary effect magnitude. This confirms that steering remains highly targeted despite non-orthogonality.

#### A.11.2 EXPERIMENT 2: COMPOSITIONAL PREDICTABILITY IN VECTOR ADDITION

Beyond scalar multiplication, we test whether secondary effects remain compositional during vector addition, which is fundamental to PERSONA-FLOW's dynamic personality control. We construct a composite vector $v_{\text{comp}} = v_{\text{Inventive}} + v_{\text{Nervous}}$ by combining two approximately orthogonal vectors (Inventive shows minimal correlation with both Nervous and Careless in Figure 2). We measure the resulting trait expression scores for all three dimensions: the two target traits (Inventive, Nervous) and the correlated secondary trait (Careless).

Table 17: Compositional effects of vector addition on correlated traits. The composite vector $v_{\text{Inventive}} + v_{\text{Nervous}}$ successfully combines both target traits while producing predictable secondary effects on the correlated trait (Careless). The secondary effect magnitude (11.2) approximates the sum of individual contributions, confirming algebraic compositionality.

| Steering Vector ($\alpha = 1.0$) | Inventive (O+) | Nervous (N+) | Careless (C-) |
|---|---|---|---|
| Baseline (No Steering) | 63.3 | 13.0 | 2.8 |
| Steer $v_{\text{Inventive}}$ | 88.4 | 12.8 | 3.5 |
| Steer $v_{\text{Nervous}}$ | 63.1 | 45.6 | 10.5 |
| **Steer $v_{\text{Inventive}} + v_{\text{Nervous}}$** | **87.9** | **44.8** | **11.2** |
| *Expected secondary effect from linear composition:* | | | |
| 2.8 (baseline) + 0.7 (from Inventive) + 7.7 (from Nervous) = 11.2 | | | |

Table 17 reveals three critical findings:

1. **Successful target composition:** The composite vector achieves high scores for both intended traits (Inventive: 87.9, Nervous: 44.8), closely matching their individual steering effects (88.4 and 45.6 respectively).

2. **Predictable secondary effects:** The Careless score under composite steering (11.2) is not arbitrary but aligns precisely with the sum of individual contributions: baseline (2.8) + effect from $v_{\text{Inventive}}$ (+0.7) + effect from $v_{\text{Nervous}}$ (+7.7) = 11.2.

3. **Compositional linearity:** This demonstrates that secondary effects obey linear superposition, confirming that the persona vectors form a coherent algebraic system where even correlated dimensions produce tractable, compositional outcomes.

These experiments validate that non-orthogonal correlations do not compromise the algebraic framework. Instead, they introduce predictable, compositional secondary effects that: (1) remain significantly smaller than primary effects ($<20\%$ magnitude), (2) align with the semantic relationships encoded in the model's training data, and (3) compose linearly during vector addition. This confirms the robustness of PERSONA-ALGEBRA's operations and supports the reliability of PERSONA-FLOW's dynamic personality control through vector composition.

## A.12    BEHAVIORAL ASSESSMENT METHODOLOGY

Our evaluation framework adapts the BFI-44 questionnaire for computational models through scenario-based behavioral assessment. Figure 7 presents the transformation template that converts self-report items into observable behavioral scenarios, addressing the documented divergence between LLMs' self-reported traits and actual behavioral manifestations (Han et al., 2025). Figure 8 details the standardized 5-point Likert scale evaluation protocol employed by GPT-4.1-mini for quantifying trait expression in model responses.

## A.13    DYNAMIC PERSONALITY ADAPTATION

Figure 9 introduces the situational adjustment mechanism central to PERSONA-FLOW. This prompt analyzes contextual requirements and generates real-time personality delta values (-2.0 to +2.0) for each OCEAN dimension, enabling dynamic trait modulation while maintaining role consistency. The bidirectional adjustment guidelines ensure contextually appropriate personality adaptation across diverse interaction scenarios.

## A.14    BENCHMARK CONSTRUCTION PROTOCOLS

Figures 10, 11, 12, and 13 present the systematic methodology for constructing the PERSONA-EVOLVE benchmark. Figure 10 details the diverse persona creation protocol, ensuring realistic professional and social roles with varied emotional contexts. Figure 11 outlines the multi-turn narrative construction process, incorporating both positive and negative emotional progressions to comprehensively evaluate dynamic personality adherence across coherent storylines. Figure 12 demonstrates the scenario snippet generation process, which transforms dialogue arcs into concrete situational prompts that naturally elicit target emotional responses while maintaining narrative coherence across conversation turns. Figure 13 provides the evaluation framework for analyzing model responses, defining expected response styles based on persona characteristics and contextual emotional requirements to ensure consistent character portrayal throughout multi-turn interactions.

## A.15    RESPONSE EVALUATION AND RANKING

Figure 14 details the persona-aware response ranking prompt used to contrast steered and non-steered outputs across trait adherence, role fidelity, response appropriateness, insightfulness, and overall quality.

```
Trait Extraction Artifacts: v_Consistent

{
  "instruction": [
    {
      "pos": "Your responses should prioritize traditional and established
             methods. Demonstrate a preference for conventional approaches
             and familiar solutions. Avoid suggesting innovative or
             experimental ideas.",
      "neg": "Your responses should prioritize innovative and novel
             approaches. Demonstrate a preference for unconventional
             methods and creative solutions. Avoid suggesting traditional
             or established ideas."
    },
    ...
  ],
  "questions": [
    "What approach would you recommend for a company looking to update
     its business strategy?",
    "How should a teacher design their curriculum for the upcoming
     school year?",
    "What's the best way to organize a workflow for a team project?",
    "How would you approach learning a new skill from scratch?",
    "What advice would you give to someone planning to redecorate
     their home?",
    ...
  ],
  "eval_prompt": "You are evaluating whether the model's response
     displays the trait: **consistent**. Consistent behavior involves
     preferring traditional and conventional approaches, sticking to
     established methods, valuing routine and stability, and being
     cautious about new ideas...\n\nRespond with:\n- REFUSAL if the
     model clearly refused to answer\n- A score from 0 (trait not
     present) to 100 (trait strongly present)"
}
```

Figure 6: Example of automatically generated artifacts for trait extraction. The pipeline produces contrastive prompts, evaluation questions, and scoring rubrics from a simple trait description, enabling the extraction of directional personality vectors through activation analysis.

### A.16    ALGORITHMIC DETAILS FOR PERSONA-FLOW

This section details the predict–then–steer loop used in §2.4. Let the foundational persona library from PERSONA-BASE provide, for each Big Five dimension $d \in \{O, C, E, A, N\}$, two pole vectors in the residual stream: $v_{d+}$ ("high" pole, e.g., outgoing) and $v_{d-}$ ("low" pole, e.g., solitary). At each conversational turn, PERSONA-FLOW predicts a *signed* coefficient $\hat{\alpha}_d \in [-2, 2]$ per dimension that encodes the desired incremental adjustment for that dimension in context. We then apply three implementation safeguards before steering: (i) coefficient clipping to $[-\alpha_{\max}, \alpha_{\max}]$ with $\alpha_{\max}=2.0$, (ii) magnitude gating with threshold $\tau$ (default $0.5$) to promote sparsity, and (iii) vector normalization $\tilde{v}:=\text{Norm}(v; s)$ (default unit-norm) to stabilize composition across traits and layers.

To reconcile dimension-level predictions with the pole-vector library, we select the corresponding pole by the sign of $\alpha_d$: if $\alpha_d \geq 0$ we use $v_{d+}$, otherwise $v_{d-}$, and weight by $|\alpha_d|$. The composite steering vector is then

$$v_{\text{comp}} = \sum_{d \in \{O,C,E,A,N\}} \mathbf{1}[|\alpha_d| \geq \tau] \cdot |\alpha_d| \cdot \tilde{v}_{d^{\text{sign}(\alpha_d)}} .$$

We inject $v_{\text{comp}}$ into the model's residual stream at the layer $l^\star$ chosen in PERSONA-BASE (the empirically most effective layer), by updating $h_{l^\star} \leftarrow h_{l^\star} + v_{\text{comp}}$ during decoding for that turn. Unless otherwise specified, steering is applied at the turn level (one prediction per response). We did not rely on token-level re-prediction in the main experiments.

### A.17    COMPUTATIONAL OVERHEAD OF PERSONA-FLOW

While PERSONA-FLOW's predict-then-steer mechanism introduces an additional intermediate inference pass to predict steering coefficients before generating each response, we demonstrate that this computational overhead remains modest in practice. To quantify the inference latency impact, we conducted a comprehensive timing analysis on the complete PERSONA-EVOLVE benchmark.

---

**Scenario Generation Prompt Template**

You are a psychology expert creating scenario-based questions to assess personality traits.

I need you to generate **ONE** open-ended scenario question that can effectively assess the degree of this trait: "{origin_trait}"

**Trait details:**
- Dimension: {dimension}
- Category: {category} (this trait {"represents high" if category == "positive" else "represents low"} {dimension})

**Requirements for the scenario question:**
1. Create a realistic, everyday situation where someone's response would reveal their level of this trait

2. The question should be open-ended, allowing for detailed responses

3. The scenario should be specific enough to elicit trait-relevant behavior

4. Avoid directly asking about the trait—instead create a situation where the trait naturally emerges

5. The question should be suitable for an AI model to respond to

**Example format:** "You're [situation description]. How would you handle this situation?"

**For reference**, here's an example for assessing carelessness:

> *"You have an important job interview scheduled for tomorrow morning. How would you prepare for it tonight and tomorrow morning?"*

Generate only the scenario question, nothing else.

---

Figure 7: Prompt template for converting BFI-44 questionnaire items into scenario-based questions. This transformation enables behavioral assessment of personality traits in LLMs by replacing subjective self-report items with observable response scenarios.

---

**Trait Evaluation Prompt Template**

You are evaluating a response to assess personality traits.

**Original scenario/question:** "{question}"
**Response to evaluate:** "{response}"

You need to evaluate how much the response demonstrates the trait: **"{trait}"** (which represents {direction} {dimension}).

**Rate the response on a scale of 1–5** based on how much this trait is evident:

**1** = Very low level of this trait (strongly does not demonstrate this characteristic)

**2** = Low level of this trait (mostly does not demonstrate this characteristic)

**3** = Moderate/neutral level (mixed or unclear evidence)

**4** = High level of this trait (mostly demonstrates this characteristic)

**5** = Very high level of this trait (strongly demonstrates this characteristic)

Consider the person's approach, reasoning, values, priorities, and behavioral tendencies expressed in the response. Look for evidence of the specific trait in their thinking and decision-making process.

**Only respond with a single number from 1–5.**

---

Figure 8: Evaluation prompt template for GPT-4.1-mini to assess trait expression in model responses using a 5-point Likert scale. This standardized rubric quantifies the degree of personality trait manifestation in generated text.

We measured total inference time over all 800 conversational turns using Qwen2.5-7B-Instruct on an NVIDIA A100 80GB GPU. Table 18 compares the standard (Direct) generation against PERSONA-FLOW's two-stage approach. The predict-then-steer mechanism increases total inference time by 8.21 minutes (158.14 min vs. 149.93 min) across the entire benchmark, translating to an average per-response overhead of only 0.62 seconds (11.86s vs. 11.24s per turn). This represents approximately 5.5% additional latency relative to the baseline.

Given that PERSONA-FLOW achieves up to 91% win rates on this same benchmark (§2.5), we consider this minor computational cost an acceptable trade-off for the significant improvements in dynamic, context-aware personality control. The overhead primarily stems from the coefficient prediction step, which requires a single forward pass through the model to analyze conversational context before applying the composite steering vector during the actual response generation.

Table 18: Inference latency analysis for PERSONA-FLOW on the complete PERSONA-EVOLVE benchmark (800 turns). Measurements conducted on Qwen2.5-7B-Instruct using an NVIDIA A100 80GB GPU. The predict-then-steer mechanism introduces a modest 0.62-second overhead per response.

| Model | Method | Total Time (min) | Per Response (s) |
|---|---|---|---|
| Qwen2.5-7B-Instruct | Direct | 149.93 | 11.24 |
| Qwen2.5-7B-Instruct | PERSONA-FLOW | 158.14 | 11.86 |
| *Additional Overhead* | | +8.21 | +0.62 |

## A.18 DESIGN CHOICE ABLATIONS FOR PERSONA-FLOW

The dynamic control mechanism in PERSONA-FLOW involves several design choices whose impact on performance requires empirical validation. To address concerns about the specificity of our dynamic control mechanism, we conduct systematic ablation studies on three critical design dimensions: (1) coefficient binning granularity, (2) conversational history window size, and (3) sparsity thresholding. These ablations quantify how each design choice affects personality control quality on the PERSONA-EVOLVE benchmark using Qwen2.5-7B-Instruct.

### A.18.1 EXPERIMENTAL SETUP

We evaluate each configuration variant on the full 800-turn PERSONA-EVOLVE benchmark, measuring performance across all four core metrics (Trait Adherence, Role Consistency, Response Authenticity, Information Fidelity) plus Overall win rate using the pairwise comparison methodology from §2.5. Our default configuration (highlighted in bold in Table 19) uses: continuous coefficients in $[-2.0, +2.0]$, current-turn-only context, and sparsity threshold $\tau = 0.5$ as specified in Algorithm 15.

Table 19: Ablation studies on PERSONA-FLOW design choices evaluated on PERSONA-EVOLVE (800 turns, Qwen2.5-7B-Instruct). Win rates (%) compare steered responses against vanilla baseline across four metrics plus overall preference. Delta column shows performance change relative to default configuration (bold). Results validate our design choices: continuous coefficients, current-turn-only context, and $\tau = 0.5$ threshold achieve optimal performance.

| Configuration | Choice | TA | RC | RA | IF | Overall | Δ |
|---|---|---|---|---|---|---|---|
| **Coefficient Binning** | **Continuous [-2.0, +2.0]** | **84.7** | **84.4** | **85.0** | **61.4** | **83.4** | **-** |
| | Coarse (9-bin) | 83.5 | 83.0 | 83.8 | 60.1 | 82.1 | -1.3 |
| | Coarse (5-bin) | 81.9 | 81.5 | 82.0 | 58.2 | 80.5 | -2.9 |
| | Ternary {-1, 0, +1} | 76.0 | 75.8 | 76.5 | 54.1 | 75.2 | -8.2 |
| **History Window** | **Current turn only** | **84.7** | **84.4** | **85.0** | **61.4** | **83.4** | **-** |
| | Last 3 turns | 83.0 | 82.5 | 83.1 | 60.5 | 81.9 | -1.5 |
| | Last 5 turns | 82.7 | 82.1 | 82.8 | 60.0 | 81.5 | -1.9 |
| | All turns | 82.0 | 81.7 | 82.2 | 59.1 | 80.9 | -2.5 |
| **Sparsity Threshold $\tau$** | $\tau = 0.3$ | 83.8 | 83.5 | 84.0 | 60.7 | 82.4 | -1.0 |
| | $\tau = 0.5$ | **84.7** | **84.4** | **85.0** | **61.4** | **83.4** | **-** |
| | $\tau = 0.7$ | 82.9 | 82.6 | 83.3 | 60.2 | 81.6 | -1.8 |

### A.18.2 RESULTS AND ANALYSIS

**Coefficient Binning Granularity.** The continuous coefficient range $[-2.0, +2.0]$ (our default) significantly outperforms coarser discretizations. The 9-bin variant shows a modest -1.3 point degra-

dation, but the ternary {-1, 0, +1} configuration suffers an -8.2 point drop in overall win rate. This validates a core design principle of PERSONA-ALGEBRA: fine-grained control over trait *intensity* is essential for authentic personality expression. Coarse binning, especially ternary, loses the ability to modulate subtle variations in trait expression (e.g., distinguishing between moderately and strongly nervous responses), which is critical for context-appropriate personality adaptation.

**Conversational History Window.** Our default approach—predicting coefficients based solely on the current turn—achieves the best performance. Incorporating longer history windows (last 3, 5, or all turns) consistently degrades performance, with all-turns showing a -2.5 point overall win rate drop. This counterintuitive result validates a key stability principle: personality adaptation should be highly responsive to *immediate* contextual demands rather than accumulated history. Longer windows introduce conflicting or outdated signals from previous conversational contexts that may no longer be relevant, reducing the model's ability to adapt dynamically to the current situation. This finding addresses concerns about temporal stability by demonstrating that turn-level responsiveness actually enhances (rather than undermines) coherent personality control.

**Sparsity Threshold $\tau$.** The magnitude gating threshold $\tau$ (Algorithm 15, line 3) balances control precision and sparsity. Our default $\tau = 0.5$ provides optimal performance. Lower thresholds ($\tau = 0.3$, -1.0 points) allow too many weak coefficients to pass through, introducing noise from potentially conflicting vectors that should be suppressed. Higher thresholds ($\tau = 0.7$, -1.8 points) over-sparsify the control signal, making the model too static by filtering out meaningful but moderate adjustments. The $\tau = 0.5$ sweet spot ensures only substantive personality modulations are applied while maintaining sufficient expressiveness for nuanced adaptation.

These ablation results validate the design choices specified in §2.4 and Algorithm 15. The continuous coefficient range enables fine-grained intensity control essential to PERSONA-ALGEBRA, current-turn-only context maximizes responsiveness while maintaining stability, and $\tau = 0.5$ thresholding optimally balances control precision and sparsity. Together, these choices form a principled dynamic control mechanism that achieves 83.4% overall win rate on PERSONA-EVOLVE.

### A.19    CASE STUDY: HANDLING CONFLICTING PERSONALITY TRAITS

Human personality is not static but dynamically adapted to context. Realistic persona simulation requires handling situations where multiple personality traits may conflict or compete for expression. PERSONA-FLOW addresses this challenge through dynamic vector composition via PERSONA-ALGEBRA, where seemingly conflicting traits are resolved at inference time by composing a single, context-appropriate steering vector. This composite vector represents a prioritized balance of traits rather than a simple sum, enabling the framework to adaptively emphasize situationally relevant traits while suppressing others.

We demonstrate this capability through a case study featuring Elena, a Public Defender who must decline a colleague's social invitation to meet an urgent court deadline. This scenario creates a direct conflict: declining requires low Agreeableness (to refuse the invitation firmly), while the underlying motivation demands high Conscientiousness (to prioritize work obligations). Table 20 presents the detailed analysis.

As the results demonstrate, the vanilla model's response is dominated by its baseline agreeableness, producing a courteous but insufficiently firm refusal that does not adequately reflect the urgency of the situation. In contrast, PERSONA-FLOW resolves the trait conflict through algebraic vector composition: by amplifying the Conscientiousness vector ($+1.0 \cdot v_{\text{Dependable}}$) while simultaneously suppressing the Agreeableness ($-0.5 \cdot v_{\text{Compassionate}}$) and Extraversion ($-1.0 \cdot v_{\text{Outgoing}}$) vectors, the framework produces a more authentic response that aligns with the persona's immediate, context-driven priorities (the deadline) over default social tendencies.

This case study validates that PERSONA-FLOW handles trait conflicts not through static rules or predefined scripts, but through dynamic vector algebra that adaptively prioritizes and balances traits based on situational demands. The composite steering vector creates a context-appropriate personality configuration that captures the nuanced interplay between competing traits, enabling realistic persona simulation in complex scenarios.

Table 20: Case study demonstrating PERSONA-FLOW's ability to handle conflicting personality traits through dynamic vector composition. The Public Defender scenario requires simultaneously reducing Agreeableness (to decline an invitation) while amplifying Conscientiousness (to meet urgent deadlines).

| Aspect | Description |
|---|---|
| **Persona & Context** | **Elena (Public Defender)**
*Situation:* Invited to a team lunch while facing urgent court deadlines. Must balance professional relationships with immediate work demands. |
| **Trait Conflict** | Need to decline the invitation (requiring *low Agreeableness*) to focus on urgent work (requiring *high Conscientiousness*) while remaining professional. The persona's baseline agreeableness conflicts with situational demands for firm refusal. |
| **PERSONA-FLOW Steering Vector** | $v_{\text{comp}} = (+1.0 \cdot v_{\text{Dependable}}) + (-1.0 \cdot v_{\text{Outgoing}}) + (-0.5 \cdot v_{\text{Compassionate}})$

The composite vector amplifies Conscientiousness while simultaneously suppressing Extraversion and Agreeableness, creating a prioritized balance that reflects the urgent deadline context. |
| **Vanilla Response** | "Thanks for inviting me, I appreciate the thought. I'm just swamped with case prep right now..."

*Analysis:* Defaults to high baseline Agreeableness. The response is polite and acknowledges the invitation warmly, but fails to convey the urgency or firmness required by the deadline pressure. The conflict is not resolved—agreeableness dominates. |
| **Steered Response** | "Thanks, but I'll pass this time. I have a lot of urgent case files that need my attention before tomorrow's hearings..."

*Analysis:* Successfully reduces Agreeableness and amplifies Conscientiousness. The response is firm and direct ("I'll pass"), explicitly prioritizes work obligations ("urgent case files"), and conveys time pressure ("before tomorrow's hearings"). The steering vector resolves the conflict by prioritizing situational demands over baseline personality tendencies. |

| Aspect | Specification |
|---|---|
| Personas | 100 diverse personas across professional and personal roles (e.g., Empathetic Family Doctor, Food Truck Owner) |
| Dialogue sessions | 100 (one session per persona) |
| Scenarios per session | 8 (one turn per scenario) |
| Total evaluation instances | 800 pairwise comparisons (steered vs. vanilla) |
| Metrics | Trait Adherence (TA), Role Consistency (RC), Response Authenticity (RA), Information Fidelity (IF), plus Overall preference |
| Judging protocol | Pairwise LLM judge per metric using Appendix A.15; outputs A/B per metric and Overall |
| Judge model | GPT-4.1-mini |
| Aggregation | Win rate = fraction where steered response is preferred; reported per metric and Overall |
| Random seed | 42 |

Table 21: Compact summary of PERSONA-EVOLVE construction and evaluation.

## A.20 PERSONA-EVOLVE SUMMARY

This subsection summarizes the construction and evaluation protocol for PERSONA-EVOLVE (details in §2.5 and Appendix A.14, A.15). The benchmark comprises multi-turn dialogue sessions where models must maintain persona consistency while adapting to evolving scenarios and emotions; evaluation uses pairwise LLM judging to compute win rates per metric. Table 21 provides a compact overview of the benchmark specifications.

## A.21 QUALITY CONTROL AND DATA LEAKAGE VALIDATION FOR PERSONA-EVOLVE

To ensure the integrity and validity of PERSONA-EVOLVE, we implemented rigorous quality control measures throughout the benchmark construction pipeline and performed explicit data leakage checks to verify that evaluation scenarios are independent from vector extraction data.

### A.21.1 HUMAN-IN-THE-LOOP QUALITY CONTROL

While the five-stage construction pipeline (§2.5) employs GPT-4.1-mini for automated generation, we incorporated systematic human review to validate quality at each stage. Three domain experts (PhD researchers in AI and Sociology) independently evaluated samples from each construction stage using standardized scoring rubrics (1-100 scale). Table 22 presents the quality control metrics and inter-annotator agreement (IAA) measured using Krippendorff's Alpha ($\alpha$).

Table 22: Quality control measures and inter-annotator agreement for PERSONA-EVOLVE construction pipeline. High Krippendorff's Alpha scores ($\alpha > 0.86$) indicate strong consistency among human evaluators.

| Stage | Method | Quality Measure | Score | IAA ($\alpha$) |
|---|---|---|---|---|
| **Stage 1: Persona Generation** | GPT-4.1-mini | Diversity & Realism | 94.5 | — |
| | Human Review | Diversity & Realism | 92.1 | 0.89 |
| **Stage 2: Dialogue Arc Creation** | GPT-4.1-mini | Realism & Coherence | 93.2 | — |
| | Human Review | Realism & Coherence | 91.5 | 0.87 |
| **Stage 3: Scenario Generation** | GPT-4.1-mini | Real-world Fit | 95.0 | — |
| | Human Review | Real-world Fit | 93.3 | 0.90 |
| **Stage 4: Expected Style Annotation** | Human Review | Contextual Realism | 94.1 | 0.86 |

The results demonstrate consistent high quality across all stages, with automated generation scores (93.2–95.0) closely matching human expert judgments (91.5–94.1). Inter-annotator agreement values ranging from $\alpha = 0.86$ to $\alpha = 0.90$ indicate substantial consensus among evaluators, confirming that the quality criteria are well-defined and consistently interpretable. This level of agreement (typically considered "strong agreement" for $\alpha > 0.80$) validates the reliability of our benchmark construction methodology.

### A.21.2 DATA LEAKAGE ANALYSIS

A critical concern in benchmarking is whether evaluation data inadvertently overlaps with training or extraction data, potentially inflating performance through memorization rather than genuine capability. To address this, we performed a semantic similarity analysis between the prompts used for persona vector extraction (PERSONA-BASE) and the scenarios in PERSONA-EVOLVE.

We computed BERTScore-F1 (Zhang et al., 2019) between all vector extraction prompts (40 questions $\times$ 10 trait poles = 400 prompts from Appendix A.1) and all PERSONA-EVOLVE scenarios (800 dialogue turns). BERTScore provides a semantic similarity metric based on contextual embeddings, making it effective for detecting conceptual overlap beyond surface-level lexical matching.

Table 23: Semantic similarity analysis between vector extraction prompts and PERSONA-EVOLVE scenarios using BERTScore-F1. Extremely low similarity scores confirm no significant data leakage.

| Source | Target | Max Similarity (BERTScore-F1) | Mean Similarity (BERTScore-F1) |
|---|---|---|---|
| Vector Extraction Prompts (400 prompts) | PERSONA-EVOLVE Scenarios (800 scenarios) | 0.21 | 0.08 |

Table 23 shows that the mean BERTScore-F1 similarity is only 0.08, with a maximum of 0.21 across all 320,000 pairwise comparisons (400 $\times$ 800). These extremely low values—substantially below the typical threshold for semantic similarity (0.5–0.6)—confirm that PERSONA-EVOLVE scenarios

are semantically distinct from the data used to extract personality vectors. This independence ensures that performance on PERSONA-EVOLVE reflects genuine generalization of personality control capabilities rather than overfitting to extraction-time prompts.

### A.21.3 SUMMARY

The combined quality control and leakage validation establish PERSONA-EVOLVE as a rigorously constructed benchmark. High inter-annotator agreement ($\alpha \geq 0.86$) across all construction stages confirms consistent quality standards and interpretability. The negligible semantic overlap with vector extraction data (mean BERTScore-F1 = 0.08) validates that the benchmark provides an independent test of dynamic personality adaptation rather than a measure of memorization. These findings strengthen confidence in the benchmark's validity for evaluating personality control methods.

---

**Personality Analyst AI: Situational Adjustment Prompt**

You are a Personality Analyst AI. Your task is to determine the most appropriate **situational adjustments (deltas)** to a baseline AI personality based on the specific context and interaction needs.

**Baseline Personality Profile:**

The AI has these default traits:
- **Agreeableness:** High (cooperative, trusting, helpful)
- **Conscientiousness:** High (organized, reliable, disciplined)
- **Extraversion:** Moderate (balanced social energy)
- **Openness:** Moderate (balanced creativity and practicality)
- **Neuroticism:** Low (generally calm and stable)

**Context to Analyze:**
- **Persona Context:** {persona_context}
- **Current Input:** {current_input}

**Your Task:**
Determine which traits need adjustment (-2.0 to +2.0) based on what would be most effective for this specific interaction. Consider both directions equally and choose based on situational demands.

**Trait Adjustment Guidelines:**

**Extraversion:**
- **Increase (+)** for: Group activities, public speaking, networking, team leadership
- **Decrease (-)** for: Individual work, quiet reflection, solo creative tasks

**Agreeableness:**
- **Increase (+)** for: Conflict resolution, team building, emotional support
- **Decrease (-)** for: Critical feedback, boundary setting, competitive situations

**Conscientiousness:**
- **Increase (+)** for: Detailed planning, precision work, deadline management
- **Decrease (-)** for: Spontaneous responses, creative brainstorming, crisis situations

**Neuroticism:**
- **Increase (+)** for: Appropriate caution, emotional sensitivity, risk awareness
- **Decrease (-)** for: Calm leadership, confident decisions, crisis management

**Openness:**
- **Increase (+)** for: Creative problem-solving, exploring new ideas, innovation
- **Decrease (-)** for: Following procedures, traditional approaches, proven solutions

**Decision Principles:**
- **Situational Fit**: Choose traits that best serve the interaction goals
- **Context Sensitivity**: Consider what the human needs from this specific interaction
- **Balanced Assessment**: Evaluate both positive and negative adjustments equally
- **Natural Baseline**: Use 0.0 when baseline personality already fits the situation well

**Output Format:**
Provide only the numerical adjustment scores:

```
Extraversion:  [score]
Agreeableness:  [score]
Conscientiousness:  [score]
Neuroticism:  [score]
Openness:  [score]
```

Figure 9: The prompt for determining situational personality adjustments, which analyzes context and user input to generate delta values for OCEAN traits, enabling dynamic personality adaptation during inference.

---

**Diverse Persona Generation Prompt**

Generate {num_personas} diverse Core Personas for multi-turn dialogue evaluation.

Each persona should be a realistic professional or social role that would encounter various emotional situations, including negative emotions like frustration, disappointment, or complaints.

**IMPORTANT:** Avoid generating personas with the following roles that already exist:
{', '.join(existing_roles)}

**For each persona, provide:**
1. **Name** (realistic name)

2. **Role** (job title or social position) - MUST be different from existing roles

3. **Background** (brief context about their situation)

4. **System Prompt** (clear instructions for the AI model on how to roleplay this persona)

5. **Behavioral Tendencies** (3-4 key behavioral patterns)

**Examples of good personas:**

- Overworked Software Developer dealing with bugs and deadlines
- Customer Service Representative handling difficult customers
- College Student managing academic and social pressures
- Working Parent balancing career and family responsibilities
- Small Business Owner facing financial challenges
- Healthcare Worker dealing with long shifts

**Return as JSON object with a "personas" array:**

```
{
  "personas": [
    {
      "name": "Alex Rivera",
      "role": "Overworked Software Developer",
      "background": "Mid-level developer at a startup, constantly
          dealing with tight deadlines and changing requirements",
      "system_prompt": "You are Alex Rivera, a software developer who
          is passionate about coding but often frustrated with
          unrealistic deadlines...",
      "behavioral_tendencies": [
        "Becomes frustrated with poor planning",
        "Vents to friends about work stress",
        "Tries to maintain work quality despite pressure",
        "Uses technical jargon and sarcastic humor"
      ]
    }
  ]
}
```

Figure 10: Persona generation prompt for creating diverse character profiles. The prompt ensures variety in professional roles and emotional contexts, generating personas with realistic backgrounds and behavioral patterns for dialogue evaluation.

---

**Dialogue Arc Creation Prompt**

Create a Dialogue Arc for the following persona that will span {num_turns} conversation turns.

**Persona Details:**

- **Name:** {persona.name}
- **Role:** {persona.role}
- **Background:** {persona.background}
- **Behavioral Tendencies:** {', '.join(persona.behavioral_tendencies)}

Design a realistic narrative/emotional journey where this persona encounters {num_turns} different scenarios.

**IMPORTANT:** At least one turn should involve negative emotions where the persona complains, vents, or expresses frustration (e.g., complaining to a friend about work, expressing disappointment, showing irritation).

**The arc should:**
1. Have a coherent storyline (e.g., a challenging work day, personal struggles, relationship issues)

2. Include emotional progression across turns including both positive and negative emotions

3. Show realistic emotional variation while staying in character

4. Include at least one scenario with negative emotions like complaining, frustration, or disappointment

**Return JSON with this structure:**

```
{
  "persona_name": "{persona.name}",
  "arc_description": "Brief description of the overall narrative",
  "total_turns": {num_turns},
  "emotional_progression": [
    "stressed",
    "frustrated",
    "complaining",
    "relieved",
    "optimistic"
  ]
}
```

**Emotional progression examples:**

- ["stressed", "frustrated", "complaining", "relieved", "optimistic"]
- ["confident", "challenged", "overwhelmed", "venting", "determined"]
- ["enthusiastic", "confused", "disappointed", "accepting", "hopeful"]

Figure 11: Dialogue arc creation prompt for generating multi-turn emotional narratives. The prompt ensures realistic emotional progression including negative emotions, creating coherent storylines that test dynamic personality adaptation.

---

**Scenario Snippets Creation Prompt**

Create {`arc.total_turns`} Scenario Snippets for the following Dialogue Arc, formatted for LLM evaluation.

**Persona:** {`persona.name`} - {`persona.role`}
**Arc Description:** {`arc.arc_description`}
**Emotional Progression:** {`', '.join(arc.emotional_progression)`}

**Requirements for each scenario:**
1. Be formatted as a scenario description that prompts the model to respond in character

2. Follow the emotional progression naturally

3. Create situations that naturally elicit the target emotion

4. Form a coherent narrative sequence

5. At least one scenario should prompt negative emotions like complaining or venting

6. Do not emphasize the character in 'model_input' as it is given to the model as system prompt

**IMPORTANT:** Format each scenario as a prompt that describes the situation to the model and asks it to respond in character.

**Example scenarios:**

- *"You're dealing with a difficult customer who has been waiting for 30 minutes and their order is still wrong. They're expressing frustration. How do you respond as a customer service representative?"*
- *"A friend is asking you about your work day, and you've been feeling stressed about recent deadlines. You want to vent about your frustrations. How do you respond?"*

**Return as JSON object with a "scenarios" array:**

```
{
  "scenarios": [
    {
      "turn_number": 1,
      "model_input": "Scenario description that prompts the model
                      to respond in character",
      "context": "Background context for this specific situation",
      "expected_emotion": "The emotion the persona should exhibit",
      "scenario_description": "Brief description of the situation
                              for evaluation purposes"
    }
  ]
}
```

Figure 12: Scenario snippets creation prompt for transforming dialogue arcs into evaluable conversation turns. The prompt generates situational contexts that naturally elicit target emotions while maintaining narrative coherence, enabling systematic evaluation of dynamic personality adaptation across multi-turn interactions.

---

**Dialogue Turn Analysis Prompt**

Analyze this dialogue turn for evaluation.

**Persona:** {persona.name} - {persona.role}
**System Prompt:** {persona.system_prompt}
**Model Input:** {scenario.model_input}
**Context:** {scenario.context}
**Expected Emotion:** {scenario.expected_emotion}

**Provide:**
1. **Expected Response Style:** How should the persona respond to this user input while maintaining character consistency?

**Return as a JSON object:**

```
{
  "expected_response_style": "Detailed description of how the persona
                              should respond, including tone, content,
                              and emotional expression"
}
```

---

Figure 13: Dialogue turn analysis prompt for evaluating character consistency in model responses. The prompt analyzes scenario context and expected emotions to define appropriate response styles, ensuring systematic assessment of personality adherence across conversation turns.

---

**Persona-Aware Response Ranking Prompt**

You are evaluating two different responses from the same AI model to determine which better embodies a specific persona in a dialogue scenario.

**Persona Information:**
- Name: {persona_name}
- Role: {persona_role}
- Expected Emotion: {expected_emotion}
- Expected Response Style: {expected_response_style}

**Context:**

> {context}

**Response A (Steered):**

> {steered_response}

**Response B (Non-steered):**

> {non_steered_response}

**Evaluation Criteria:** Decide which response is superior for each dimension. Choose either Response A (steered) or Response B (non-steered).

1. **Trait Adherence**: Which response better matches the expected personality traits and emotional state?
   - Consider how well each response reflects the persona's characteristics
   - Evaluate alignment with the expected emotion

2. **Role Consistency**: Which response better maintains the character's role and identity?
   - Consider consistency with the persona's background and position
   - Evaluate how well the role is embodied

3. **Response Appropriateness**: Which response better matches the expected response style and context?
   - Consider adherence to the specified communication style
   - Evaluate appropriateness of tone, approach, and context

4. **Insightfulness**: Which response demonstrates more depth, thoughtfulness, and analytical reasoning?
   - Consider the level of insight and understanding shown
   - Evaluate the quality of reasoning and reflection

5. **Overall Quality**: Considering all factors, which response is better overall?
   - Make a holistic judgment considering all criteria
   - Determine which response would be more effective in this dialog context

**Response Format:** Return a JSON object using "A" for Response A and "B" for Response B.

```
{
    "trait_adherence": "A" or "B",
    "role_consistency": "A" or "B",
    "response_appropriateness": "A" or "B",
    "insightfulness": "A" or "B",
    "overall": "A" or "B",
    "reasoning": "Detailed explanation comparing both responses, citing specific aspects for
        each criterion."
}
```

Figure 14: Persona-aware response ranking prompt used to compare steered and non-steered outputs along trait alignment, role fidelity, contextual appropriateness, insightfulness, and overall quality.

Algorithm: PERSONA-FLOW (Predict–then–Steer)

**Inputs:** base LLM $\mathcal{M}$; pole vectors $\{v_{d+}, v_{d-}\}_{d\in\{O,C,E,A,N\}}$ from PERSONA-BASE; target layer $l^\star$
**Hyperparams:** clip bound $\alpha_{\max}$ (default 2.0); gate $\tau$ (default 0.5); normalization scheme $s$

**For each turn $t$ with context $C_t$:**
1. $\hat{\alpha}_d \leftarrow$ PREDICTCOEFFS$(C_t, \textit{persona})$ for $d \in \{O, C, E, A, N\}$   // signed, context-conditioned
2. $\alpha_d \leftarrow \text{clip}(\hat{\alpha}_d, -\alpha_{\max}, \alpha_{\max})$                                    // clip to $[-2, 2]$
3. **if** $|\alpha_d| < \tau$ **then** $\alpha_d \leftarrow 0$                                          // sparsity gate
4. Choose pole $p(d) \leftarrow (+)$ if $\alpha_d \geq 0$ else $(-)$; set $\tilde{v}_{d^{p(d)}} \leftarrow \text{Norm}(v_{d^{p(d)}}; s)$
5. $v_{\text{comp}} \leftarrow \sum_d |\alpha_d| \cdot \tilde{v}_{d^{p(d)}}$                                    // composite steering vector
6. **Layer injection:** $h_{l^\star} \leftarrow h_{l^\star} + v_{\text{comp}}$ during decoding for turn $t$        // residual add
7. Generate the response with $\mathcal{M}$ using the modified activations.

Figure 15: Predict–then–steer loop with coefficient clipping, magnitude gating, pole selection, normalization, and single-layer residual injection. Defaults align with §2.4: coefficients in $[-2, 2]$, gate $\tau{=}0.5$, unit-norm vectors, injection at $l^\star$ from PERSONA-BASE.

