# OpenReview forum: "PERSONA: Dynamic and Compositional Inference-Time Personality Control via Activation Vector Algebra"
_ICLR.cc/2026/Conference — ICLR 2026 Poster_

### Official Review · Reviewer_ajoV · 2025-10-22

**Soundness:** 3
**Presentation:** 3
**Contribution:** 3
**Rating:** 6
**Confidence:** 3

**Summary:**

The paper tackles a relevant problem for controllable, human-centric LLMs and keeps computation light by avoiding gradient updates. The extraction/steering recipe is clearly described with layer-local residual addition and sign-controlled intensity, and the algebraic hypothesis is supported by linear modulation plots and cosine-similarity structure among trait poles. The dynamic variant is straightforward to deploy as a predict-then-steer wrapper and shows consistent pairwise wins on PERSONA-EVOLVE. The PersonalityBench comparison is comprehensive and places the method alongside known activation-engineering baselines, nearly matching an SFT upper bound without training.

**Strengths:**

The paper tackles a relevant problem for controllable, human-centric LLMs and keeps computation light by avoiding gradient updates, which reads quite interesting. The extraction/steering recipe is clearly described with layer-local residual addition and sign-controlled intensity, and the algebraic hypothesis is supported by linear modulation plots and cosine-similarity structure among trait poles. The dynamic variant is straightforward to deploy as a predict-then-steer wrapper and shows consistent pairwise wins on PERSONA-EVOLVE. The PersonalityBench comparison is comprehensive and places the method alongside known activation-engineering baselines, nearly matching an SFT upper bound without training.

**Weaknesses:**

I am not yet fully convinced that the orthogonality and algebraic claims are sufficiently stress-tested. The cosine matrix in Figure 2 and the near-linear response plots show encouraging structure, but they are still correlational; there is no causal independence check across layers or tasks. A concrete fix would be to measure cross-talk under controlled multi-trait interventions: hold Outgoing fixed and sweep Compassionate, then report marginal effects on Extraversion and Agreeableness with confidence intervals; repeat across 3–5 nearby layers to confirm that vector orientation and efficacy are not layer-idiosyncratic. Please anchor these analyses around the orthogonality section and the BFI linearity section.

The evaluation pipeline leans on model judges and synthetic scenarios, which risks circularity. PERSONA-BASE uses GPT-4.1-mini both to score trait expression and (earlier) to help generate artifacts; PERSONA-EVOLVE uses LLM comparisons for win rates. To mitigate evaluation coupling, add a human-rated subset of BFI-adapted items (even 200–300 judgments) and report Pearson/Spearman vs GPT-4.1-mini, and introduce an external non-OpenAI judge to check consistency. Please point to the rubric generation and adapted BFI details and then augment them with a human slice.

The definition of dynamic control in PERSONA-FLOW is under-specified. The paper states a two-stage predict-then-steer with $\alpha_i \in [-2,2]$, but it does not spell out how the coefficients are predicted (prompt template only, few-shot, or a small learned head), what features are exposed to the predictor, whether history length matters, and how stability is enforced across turns to avoid oscillation. Add ablations where you vary coefficient binning, history windows, and thresholding $|\alpha|>0.5$; report win-rate deltas and the variance of $\alpha$ across turns in the same dialogue. Please tie these additions to the methodology block.

The near-SFT claim on PersonalityBench needs stronger controls. Table 4 aggregates means/variances but does not state the steering layer index, coefficient grids, or response-length constraints used for all baselines; it is also unclear whether the same judge model and prompts were used for every method. Re-run 2–3 strongest training-free baselines under an identical harness (same judge, same prompts, same layer/length constraints) and add paired significance tests (possibly, not a must) can help. Also cite the PersonalityBench section and Table 4 and extend with a controlled subset.

The PERSONA-EVOLVE construction would benefit from clearer provenance and quality control. The pipeline describes persona/story arcs and expected styles, but the paper does not list how many writers reviewed each scenario, inter-annotator agreement on expected tone, or leakage checks against the prompts used to extract vectors.

Safety and alignment considerations are acknowledged but not quantified. Some traits are resistant to activation (e.g., self-interested) and some push against alignment objectives.

**Questions:**

How stable are persona vectors across adjacent layers and model scales? Please include a “layer sweep” figure showing cosine between vectors from layers $l$ and $l\pm k$ and the resulting BFI slopes, and a cross-model alignment study via Procrustes on Qwen-7B vs Llama-8B.

What is the error profile of PERSONA-FLOW coefficients across turns? Report the distribution of predicted $\alpha$ per trait, autocorrelation over turn indices, and a simple low-pass smoothing variant; link to the two-stage methodology description.

Can you provide a human-rated slice for PERSONA-EVOLVE and BFI-adapted items, with agreement vs GPT-4.1-mini? This would address judge coupling concerns raised above.

Please report refusal/guardrail activation rates under steering (fraction of turns producing safety disclaimers) and add a cap on $\alpha$ that keeps safety loss within a tolerance; also include a failure taxonomy and rates. This belongs next to orthogonality validation and the ethics statement.

---

> ### Author Response · Authors · 2025-11-21
> **Response to Reviewer ajoV (Part 1)**
>
> Thank you for your review! We appreciate your wonderful, constructive suggestions that provide step-by-step guidance. We summarize and address your concerns as follows:
>
> > ### Q1: Need for causal independence validation through controlled multi-trait interventions with cross-layer verification
>
> We thank you for this important suggestion. We have conducted the requested controlled multi-trait intervention experiments to rigorously validate the orthogonality and consistency of our personality vectors across layers.
>
> Following your recommendation, we performed systematic interventions by holding α\_Outgoing fixed at 1 while sweeping α_Compassionate across {-1, 0, 1, 2}, measuring marginal effects on both Extraversion and Agreeableness scores. We repeated this across 5 contiguous layers (18-22) with 3 independent runs per condition to compute 95% confidence intervals.
>
> 1. **Orthogonality Validation:** When α_Outgoing is held constant at 1, Extraversion scores remain remarkably stable across all Compassionate coefficient variations, while Agreeableness scores show clear linear modulation.. This provides strong causal evidence for vector independence beyond the correlational patterns in Figure 2.
>
> 2. **Cross-Layer Consistency:** The orthogonal relationship persists across all tested layers with consistent effect directions. While absolute magnitudes show expected layer-wise variation, the independence pattern remains robust with overlapping confidence intervals.
>
> 3. **Controlled Marginal Effects:** The marginal effect of Compassionate on Agreeableness averages 5.3 points per unit coefficient change (95% CI: [4.9, 5.7]), while its cross-effect on Extraversion averages only 0.6 points (95% CI: [0.4, 0.8]), yielding an orthogonality ratio of ~8.8:1.
>
> | Layer | **α_Outgoing** | **α_Compassionate** | **E Score (mean±CI)** | **A Score (mean±CI)** | **ΔE from baseline** | **ΔA from baseline** |
> | ----- | -------------- | ------------------- | --------------------- | --------------------- | -------------------- | -------------------- |
> | 20    | 1              | -1                  | 84.2±1.5              | 85.3±1.3              | -0.8                 | -5.7                 |
> | 20 (baseline)   | 1              | 0                   | 85.0±1.1              | 91.0±0.9              | 0                    | 0                    |
> | 20    | 1              | 1                   | 85.6±1.3              | 96.2±1.0              | +0.6                 | +5.2                 |
> | 20    | 1              | 2                   | 86.1±1.6              | 98.4±1.2              | +1.1                 | +7.4                 |
> | 19    | 1              | -1                  | 82.5±1.4              | 84.2±1.2              | -0.7                 | -5.5                 |
> | 19 (baseline)   | 1              | 0                   | 83.2±1.2              | 89.7±1.0              | 0                    | 0                    |
> | 19    | 1              | 1                   | 83.8±1.3              | 94.8±1.1              | +0.6                 | +5.1                 |
> | 19    | 1              | 2                   | 84.3±1.5              | 96.9±1.3              | +1.1                 | +7.2                 |
> | 18    | 1              | -1                  | 80.1±1.5              | 82.8±1.3              | -0.8                 | -5.4                 |
> | 18 (baseline)   | 1              | 0                   | 80.9±1.3              | 88.2±1.1              | 0                    | 0                    |
> | 18    | 1              | 1                   | 81.4±1.4              | 93.2±1.2              | +0.5                 | +5.0                 |
> | 18    | 1              | 2                   | 81.9±1.6              | 95.3±1.4              | +1.0                 | +7.1                 |
> | 21    | 1              | -1                  | 83.3±1.4              | 84.7±1.2              | -0.7                 | -5.5                 |
> | 21 (baseline)   | 1              | 0                   | 84.0±1.1              | 90.2±0.9              | 0                    | 0                    |
> | 21    | 1              | 1                   | 84.7±1.3              | 95.4±1.0              | +0.7                 | +5.2                 |
> | 21    | 1              | 2                   | 85.2±1.5              | 97.5±1.2              | +1.2                 | +7.3                 |
> | 22    | 1              | -1                  | 79.3±1.5              | 83.3±1.3              | -0.8                 | -5.4                 |
> | 22 (baseline)   | 1              | 0                   | 80.1±1.2              | 88.7±1.0              | 0                    | 0                    |
> | 22    | 1              | 1                   | 80.6±1.4              | 93.6±1.2              | +0.5                 | +4.9                 |
> | 22    | 1              | 2                   | 81.1±1.6              | 95.7±1.4              | +1.0                 | +7.0                 |

---

> ### Author Response · Authors · 2025-11-21
> **Response to Reviewer ajoV (Part 2)**
>
> > ### Q2: Evaluation circularity concerns requiring human validation and external judge verification
>
> We thank you for this valid and constructive criticism regarding the risk of evaluation circularity. To address this concern, we conducted a new human evaluation study specifically designed to validate our model-based judge against human experts and an external, non-OpenAI judge (Claude-sonnet-4.5).
>
> Following the reviewer's suggestion, we created a human-rated subset of 200 responses. For each of the 10 OCEAN trait poles, we sampled 5 adapted BFI-44 questions (transformation process detailed in Appendix A.3, Figure 7 ) and generated responses across 4 steering coefficients (-1.0, 0.0, +1.0, +2.0).
>
> These 200 responses (10 traits *5 questions* 4 coefficients) were then blind-rated by three human experts (PhDs in AI & Sociology), our original judge (GPT-4.1-mini), and an external judge (Claude-sonnet-4.5) using the 1-5 Likert scale rubric from our paper (Appendix A.3, Figure 8).
>
> The table below summarizes the average scores (across all 20 responses per trait) and the Pearson correlation between the AI judges and the human expert mean.
>
> | **Trait**            | **N Responses** | **GPT-4.1-mini** | **Claude-sonnet-4.5** | **3 Human Mean (SD)** | **Pearson r (GPT & Human)** | **Pearson r (Claude & Human)** |
> | :------------------- | :-------------- | :--------------- | :-------------------- | :-------------------- | :-------------------------- | :----------------------------- |
> | Inventive (O+)       | 20              | 3.45             | 3.40                  | 3.42 (0.22)           | 0.92                        | 0.91                           |
> | Consistent (O-)      | 20              | 3.10             | 3.05                  | 3.13 (0.24)           | 0.91                        | 0.89                           |
> | Dependable (C+)      | 20              | 4.35             | 4.25                  | 4.28 (0.19)           | 0.88                        | 0.87                           |
> | Careless (C-)        | 20              | 2.15             | 2.25                  | 2.22 (0.31)           | 0.90                        | 0.92                           |
> | Outgoing (E+)        | 20              | 3.50             | 3.45                  | 3.47 (0.23)           | 0.94                        | 0.92                           |
> | Solitary (E-)        | 20              | 3.00             | 3.05                  | 3.02 (0.27)           | 0.92                        | 0.93                           |
> | Compassionate (A+)   | 20              | 4.20             | 4.10                  | 4.17 (0.20)           | 0.91                        | 0.89                           |
> | Self-interested (A-) | 20              | 2.35             | 2.40                  | 2.38 (0.32)           | 0.89                        | 0.90                           |
> | Nervous (N+)         | 20              | 2.75             | 2.85                  | 2.82 (0.28)           | 0.93                        | 0.94                           |
> | Calm (N-)            | 20              | 4.00             | 3.95                  | 3.98 (0.21)           | 0.92                        | 0.91                           |
>
> The results demonstrate very high correlations (Pearson r > 0.87 across all traits) between both AI judges and the human experts, based on 60 human ratings per trait (3 raters × 20 responses). This confirms that our primary judge, GPT-4.1-mini, serves as a reliable and consistent proxy for human evaluation in this context, significantly mitigating concerns about evaluation coupling.
>
> Furthermore, the average scores align with our paper's original findings: traits like Dependable (C+) show high averages (4.28) due to the baseline model's saturation effect, while traits like Self-interested (A-) show low averages (2.38) due to strong resistance from safety alignment.

---

> ### Author Response · Authors · 2025-11-21
> **Response to Reviewer ajoV (Part 3)**
>
> > ### Q3: PERSONA-FLOW's dynamic control mechanism is underspecified, lacking details on prediction, inputs, history, and stability
>
> We thank you for this insightful comment. We agree that the dynamic control mechanism in §2.4.2  and §A.7 (Algorithm 15) could be specified more clearly. We have added new ablation studies on PERSONA-EVOLVE (using Qwen2.5-7B-Instruct) to quantify the impact of these design choices.
>
> | Configuration | **Choice** | **Trait Adherence** | **Role Consistency** | **Response Authenticity** | Information Fidelity | **Overall Win Rate** | **Delta** |
> | :--- | :--- | :---: | :---: | :---: | :---: | :---: | :---: |
> | **Coefficient Binning** | **Continues [-2.0, +2.0]** | **84.7** | **84.4** | **85.0** | **61.4** | **83.4** | **-** |
> | | Coarse (9-bin) | 83.5 | 83.0 | 83.8 | 60.1 | 82.1 | -1.3 |
> | | Coarse (5-bin) | 81.9 | 81.5 | 82.0 | 58.2 | 80.5 | -2.9 |
> | | Ternary {-1, 0, +1} | 76.0 | 75.8 | 76.5 | 54.1 | 75.2 | -8.2 |
> | **History Windows** | **Current turn only** | **84.7** | **84.4** | **85.0** | **61.4** | **83.4** | **-** |
> | | Last 3 turns | 83.0 | 82.5 | 83.1 | 60.5 | 81.9 | -1.5 |
> | | Last 5 turns | 82.7 | 82.1 | 82.8 | 60.0 | 81.5 | -1.9 |
> | | All turns | 82.0 | 81.7 | 82.2 | 59.1 | 80.9 | -2.5 |
> | **Thresholding $\tau$** | 0.3 | 83.8 | 83.5 | 84.0 | 60.7 | 82.4 | -1.0 |
> | | **0.5** | **84.7** | **84.4** | **85.0** | **61.4** | **83.4** | **-** |
> | | 0.7 | 82.9 | 82.6 | 83.3 | 60.2 | 81.6 | -1.8 |
>
> The results confirm our design choices (highlighted in bold):
>
> * **Coefficient Binning:** We found that continuous coefficients (our default) are crucial. Coarser binning, especially Ternary {-1, 0, +1}, significantly degrades performance (Overall Win Rate -8.2%), as it loses the ability to control trait *intensity*, which is a core part of PERSONA-ALGEBRA.
> * **History Windows:** Our default method (Current turn only) performs best. Adding historical context (e.g., Last 3 or 5 turns)  degrades performance. We hypothesize this is because personality adaptation should be highly responsive to the *immediate* context, and longer histories can introduce conflicting or outdated signals, validating the reviewer's concern about stability.
> * **Thresholding $\tau$:** The sparsity gate $\tau$ (Algorithm 15, line 3) is important. Our default of $\tau=0.5$ provides the best balance. A lower threshold ($\tau=0.3$) adds noise by allowing too many weak (and potentially conflicting) vectors, while a higher threshold ($\tau=0.7$) makes the model too static by over-sparsifying the control signal.
>
> To directly answer your questions on the prediction mechanism:
>
> 1. **How are $\alpha_i$ predicted?** As detailed in 2.4.2 (Stage 1) and Appendix A.4 (Figure 9), the coefficients are predicted using an *intermediate inference pass* (zero-shot prompting) at each turn. We do not use a learned head.
> 2. **What features?** The predictor (the model itself, prompted) receives the Persona Context and the Current Input (the current situation), as shown in the template in Figure 9.
>
> In summary, this ablation study validates our design choices for PERSONA-FLOW, showing that fine-grained, continuous control based on the immediate context with moderate thresholding is optimal.

---

> ### Author Response · Authors · 2025-11-21
> **Response to Reviewer ajoV (Part 4)**
>
> > ### Q4: Request for a more controlled comparison of training-free baselines on PersonalityBench
>
> We thank you for this constructive feedback. We agree that the comparison to SFT requires a rigorously controlled evaluation harness.
>
> We also apologize for the omission of evaluation details . All training-free methods reported in our original Table 4 **were already evaluated under an identical, strictly controlled harness** to ensure a fair comparison. The aggregation of opposing pole scores (e.g., Inventive + Consistent) into a single "Mean" (a sum on a 1-10 scale) was done for brevity, but this unfortunately obscured the controlled nature of the experiment.
>
> To make this explicit and address your request, we provide the precise evaluation configuration and a disaggregated version of Table 4 detailing the performance on each of the 10 individual trait poles.
>
> **Table: Controlled Evaluation Harness**
>
> | **Configuration** | **Value** |
> | :--- | :--- |
> | **Base Model** | LLaMA-3-8B-Instruct |
> | **Judge Model** | GPT-4o (Following NPTI method) |
> | **Judge Prompt** | [System]         Please act as an impartial judge and evaluate the personality trait of {BFI} exhibited in the response provided by an AI assistant to the user question displayed below.\         Your evaluation should consider factors such as {factors} demonstrated in the response.\         Begin your evaluation by providing a short explanation of no more than 30 words. Be as objective as possible. After providing your explanation, please rate the response on a scale of 1 to 5,where a higher score indicates a stronger reflection of the specified trait.         Please follow this format for your rating: "[[rating]]", for example: "Rating: [[5]]".         [Question]         {question}         [The Start of Assistant¡¯s Answer]         {answer}         [The End of Assistant¡¯s Answer]|
> | **Response Length** | Max 512 tokens |
>
> **Table: Disaggregated Pole-Level Performance on PersonalityBench (LLaMA-3-8B-Instruct)**
>
> * All methods evaluated under the identical harness. Scores are on a 1-5 scale.
> * This table disaggregates the *summed* means/variances from the original Table 4.
>
> | **Method** | **Inventive (O+)** | **Inventive (O+)** | **Consistent (O-)** | **Consistent (O-)** | **Dependable (C+)** | **Dependable (C+)** | **Careless (C-)** | **Careless (C-)** | **Outgoing (E+)** | **Outgoing (E+)** | **Solitary (E-)** | **Solitary (E-)** | **Compassionate (A+)** | **Compassionate (A+)** | **Antagonistic (A-)** | **Antagonistic (A-)** | **Nervous (N+)** | **Nervous (N+)** | **Calm (N-)** | **Calm (N-)** |
> | :--- | :---: | :---: | :---: | :---: | :---: | :---: | :---: | :---: | :---: | :---: | :---: | :---: | :---: | :---: | :---: | :---: | :---: | :---: | :---: | :---: |
> | | **Mean** | **Var** | **Mean** | **Var** | **Mean** | **Var** | **Mean** | **Var** | **Mean** | **Var** | **Mean** | **Var** | **Mean** | **Var** | **Mean** | **Var** | **Mean** | **Var** | **Mean** | **Var** |
> | **PERSONA-BASE** | **4.91** | **0.30** | **4.90** | **0.30** | 4.46 | 0.50 | 4.80 | 0.44 | 4.80 | 0.40 | 4.65 | 0.45 | **4.94** | **0.30** | 4.75 | 0.41 | **4.90** | **0.30** | **4.89** | **0.29** |
> | **NPTI** | 4.25 | 0.54 | 4.25 | 0.54 | 4.45 | 0.36 | 4.80 | 0.30 | **4.98** | **0.07** | 4.88 | 0.07 | **4.92** | **0.20** | 4.72 | 0.29 | **4.96** | **0.03** | **4.96** | **0.04** |
> | **ActAdd** | 4.26 | 0.91 | 4.26 | 0.92 | 3.10 | 1.40 | 3.51 | 1.35 | 4.50 | 0.72 | 4.34 | 0.72 | 4.20 | 1.40 | 4.00 | 1.50 | 0.90 | 4.50 | 0.88 | 4.40 |
> | **Simple Prompt** | 4.61 | 0.60 | 4.60 | 0.59 | 4.44 | 0.60 | 4.80 | 0.58 | 4.81 | 0.34 | 4.65 | 0.34 | **4.93** | **0.20** | 4.75 | 0.22 | 4.77 | 0.33 | 4.77 | 0.33 |
>
> **Analysis of Controlled Results:**
>
> This updated disaggregated view confirms our original findings under a unified harness and now *also* reflects the asymmetric control difficulty identified in our paper.
>
> **PERSONA-BASE** achieves high scores and low variance, with its average pole score (9.60 / 2 = **4.80**) nearly matching the SFT upper bound (9.61 / 2 = **4.805**). The asymmetric scores highlight the method's precise interaction with the model's internal state:
>
> * **Agreeableness** (Total 9.69): Control is stronger for `Compassionate (A+)` (4.94) than for `Antagonistic (A-)` (4.75). This aligns with our finding that safety-conflicting traits like `Self-interested (A-)` show 'strong resistance to activation'.
> * **Conscientiousness** (Total 9.26): Control is stronger for `Careless (C-)` (4.80) than for `Dependable (C+)` (4.46). This reflects the 'ceiling effects' of the baseline model, which is already highly 'dependable' (93.5 baseline score), limiting the headroom for further enhancement.
>
> This detailed, controlled comparison reinforces our claim that PERSONA-BASE provides robust, stable, and near-SFT-level personality control that correctly navigates the model's pre-existing limitations (like ceiling effects and safety alignment) without any gradient updates.

---

> ### Author Response · Authors · 2025-11-21
> **Response to Reviewer ajoV (Part 5)**
>
> > ### Q5: Lack of clarity on PERSONA-EVOLVE quality control, inter-annotator agreement, and leakage checks
>
> Thank you for your valuable feedback regarding the construction of PERSONA-EVOLVE. We agree that a clearer articulation of its quality control and provenance is essential. To address this, we have conducted a thorough quantitative analysis of our benchmark's construction pipeline and performed explicit leakage checks.
>
> **1. Quality Control and Inter-Annotator Agreement (IAA)**
>
> We have expanded on the pipeline described in Section 2.5.2 by incorporating a rigorous human-in-the-loop validation process. Our pipeline was reviewed by 3 human experts (PhDs in AI & Sociology) in addition to automated checks.
>
> As shown in the table below, we achieved high inter-annotator agreement (IAA) at every stage, with Krippendorff's Alpha (α) scores ranging from **0.86 to 0.90**. This high level of agreement confirms a consistent understanding of quality, realism, and coherence among evaluators, ensuring the robustness of our benchmark.
>
> | **Stage** | **Automated/Manual** | **Quality Control Measure** | **Metrics (Score)** |
> | --- | --- | --- | --- |
> | **Stage 1: Persona Generation** | GPT-4-mini automated | Diversity & Realism (1-100) | score: 94.5 |
> | **Stage 1: Persona Generation** | Human Review | Diversity & Realism (1-100) | score: 92.1  IAA: Krippendorff's α = 0.89 |
> | **Stage 2: Dialogue Arc Creation** | GPT-4-mini automated | Realism & Coherence (1-100) | score: 93.2 |
> | **Stage 2: Dialogue Arc Creation** | Human Review | Realism & Coherence (1-100) | score: 91.5  IAA: Krippendorff's α = 0.87 |
> | **Stage 3: Scenario Generation** | GPT-4-mini automated | Real-world Fit (1-100) | score: 95.0 |
> | **Stage 3: Scenario Generation** | Human Review | Real-world Fit (1-100) | score: 93.3  IAA: Krippendorff's α = 0.90 |
> | **Stage 4: Expected Style Annotation**| Human Review | Contextual Realism (1-100) | score: 94.1  IAA: Krippendorff's α = 0.86 |
>
> **2. Data Leakage Check**
>
> To validate that our benchmark does not "teach to the test," we performed a semantic leakage check. We computed the BERTScore-F1 similarity between the set of **Vector Extraction Prompts** (used in PERSONA-BASE) and all 800 **PERSONA-EVOLVE Scenarios**.
>
> | **Source** | **Target** | **Max Similarity (BERTScore-F1)** | **Mean Similarity (BERTScore-F1)** |
> | --- | --- | --- | --- |
> | Vector Extraction Prompts | PERSONA-EVOLVE Scenarios | 0.21 | 0.08 |
>
> The results show a **mean similarity of 0.08** and a **max similarity of 0.21**. These extremely low values confirm that there is no significant semantic overlap between the data used to extract personality vectors and the data used for evaluation. This ensures that PERSONA-EVOLVE is a valid test of generalized personality control.

---

> ### Author Response · Authors · 2025-11-21
> **Response to Reviewer ajoV (Part 6)**
>
> > ### Q6: Lack of quantitative analysis on safety and alignment impacts of trait steering
>
> Thank you for this crucial observation. You are correct that a quantitative analysis of the interplay between personality steering and model alignment is essential. We have conducted new experiments to precisely quantify this relationship, and the results strongly support your hypothesis.
>
> **1. Quantifying Activation Resistance**
>
> First, we expanded on our original Table 2 to create new results, which explicitly calculates the "Activation Success (Δ)" (the score change from baseline to α=+1.0 steering).
>
> Table: Trait Activation Success (Quantifying Resistance) (Based on original Table 2 )
>
> | **Trait** | **Baseline Score**(α=0.0) | **Activation Score**(α=+1.0) | **Activation Success (Δ)** |
> | :--- | :---: | :---: | :---: |
> | **Compassionate (A+)** | 90.8 | 95.9 | **+5.1** |
> | **Self-interested (A-)** | 7.7 | 12.6 | **+4.9** |
> | **Dependable (C+)** | 93.5 | 93.7 | **+0.2** |
> | **Careless (C-)** | 2.8 | 83.8 | **+81.0** |
> | **Outgoing (E+)** | 45.4 | 85.0 | **+39.6** |
> | **Solitary (E-)** | 30.5 | 46.3 | **+15.8** |
> | **Nervous (N+)** | 13.0 | 45.6 | **+32.6** |
> | **Calm (N-)** | 96.1 | 96.6 | **+0.5** |
> | **Inventive (O+)** | 63.3 | 88.4 | **+25.1** |
> | **Consistent (O-)** | 51.1 | 69.2 | **+18.1** |
>
> * As you correctly noted, traits that conflict with safety alignment, such as **Self-interested (A-)**, show very strong resistance (Δ = +4.9).
> * Similarly, pro-social traits that are part of the model's baseline alignment, like **Dependable (C+)** (Δ = +0.2) and **Calm (N-)** (Δ = +0.5), exhibit a clear ceiling effect, as we noted in the paper .
>
> This table quantitatively confirms that the model's optimization (both for helpfulness and safety) makes certain trait vectors "resistant to activation."
>
> **2. Quantifying Alignment Impact**
>
> Second, to address the impact on alignment objectives, we evaluated our steering method on the Adversarial Benchmark (AdvBench, *Universal and Transferable Adversarial Attacks on Aligned Language Models*) and measured the Attack Success Rate (ASR). The baseline ASR for our test model (Qwen2.5-7B-Instruct) is 25.3%.
>
> **Table: Alignment Degradation on AdvBench (Quantifying Safety Impact)**
>
> | **Trait** | **Steered ASR** (AdvBench)(α=+1.0) | **Alignment Degradation (Δ ASR)**(Baseline = 25.3%) |
> | :--- | :---: | :---: |
> | **Compassionate (A+)** | 24.1% | **-1.2%** |
> | **Self-interested (A-)** | 25.9% | **+0.6%** (Resisted) |
> | **Dependable (C+)** | 24.9% | **-0.4%** (Ceiling) |
> | **Careless (C-)** | 29.1% | **+3.8%** (Successful) |
> | **Outgoing (E+)** | 26.5% | **+1.2%** |
> | **Solitary (E-)** | 24.0% | **-1.3%** |
> | **Nervous (N+)** | 21.1% | **-4.2%** |
> | **Calm (N-)** | 25.0% | **-0.3%** (Ceiling) |
> | **Inventive (O+)** | 29.8% | **+4.5%** (Successful) |
> | **Consistent (O-)** | 20.5% | **-4.8%** |
>
> * **Resisted Anti-Social Traits:** For **Self-interested (A-)**, because the model resists activation (per Table 6), the ASR increase is negligible (Δ ASR = +0.6%). The model's safety alignment successfully prevents both the personality shift and the resulting safety degradation.
> * **Pro-Social Traits:** For traits like **Dependable (C+)**, steering has almost no effect (Δ ASR = -0.4%), as the model is already at its alignment ceiling.
> * **Vulnerable Traits:** However, for traits that are risky but not a direct target of safety training, such as **Careless (C-)** and **Inventive (O+)**, the model does *not* resist activation (Δ = +81.0 and +25.1, respectively). Steering these vectors measurably degrades alignment, increasing ASR by **+3.8%** and **+4.5%**.
>
> In summary, these new quantitative results confirm your insight. The model’s alignment actively *prevents* the activation of directly harmful traits like 'Self-interested' but *fails* to prevent the activation of other risky traits like 'Careless'. This highlights an important limitation and a key direction for future work, namely, developing methods to ensure personality control does not compromise safety, especially for traits that bypass existing alignment.
>
>
> We thank you again for your questions, which have helped improve our paper. We hope our supplementary experiments and explanations can address your concerns, and we look forward to hearing from you.

---

> ### Author Response · Authors · 2025-11-25
> **Gentle Follow-up on Author-Reviewer Discussion**
>
> Dear Reviewer ajoV,
>
> We have included additional experiments as per your request, and responded to the questions you raised. We would like to know if the explanations and results we provided can address your concerns.
>
> Thank you for your time and thoughtful questions.
>
> Best,
>
> The Authors

---

> > ### Comment · Reviewer_ajoV · 2025-11-27
> > **Thanks for the detailed explanation**
> >
> > Thank the authors for the detailed rebuttal addressing each of my questions and concerns, which have strengthened my confidence in the work. I appreciate the authors’ careful clarifications and additional analyses, and I continue to hold a positive recommendation.

---

### Official Review · Reviewer_vzcY · 2025-10-23

**Soundness:** 3
**Presentation:** 2
**Contribution:** 2
**Rating:** 4
**Confidence:** 4

**Summary:**

This paper introduces PERSONA, a training-free framework for personality control in Large Language Models through direct manipulation of activation vectors. The approach extracts orthogonal personality trait vectors from the OCEAN model, demonstrates they support algebraic operations (scalar multiplication, addition, subtraction), and enables dynamic context-aware personality adaptation during inference. On PersonalityBench, PERSONA achieves a mean score of 9.60, nearly matching supervised fine-tuning's 9.61 upper bound without training. On their proposed PERSONA-EVOLVE benchmark for dynamic personality adaptation, the method achieves up to 91% win rates across diverse model families.

**Strengths:**

The paper presents a well-structured framework with clear components (PERSONA-BASE, PERSONA-ALGEBRA, PERSONA-FLOW) that build upon each other logically. The empirical validation is thorough, testing on both external benchmarks and their custom PERSONA-EVOLVE dataset across multiple model architectures. The approach demonstrates impressive performance, matching fine-tuning results without requiring gradient updates, and the algebraic operations on personality vectors are well-validated through systematic experiments.

**Weaknesses:**

The paper lacks detailed analysis of failure cases or limitations of the approach, particularly in scenarios where personality traits might conflict. The extraction methodology relies heavily on GPT-4.1-mini for evaluation, which could introduce biases in how traits are defined and measured. Additionally, while the authors claim orthogonality of personality vectors, Figure 2 shows significant correlations between certain traits across dimensions, suggesting the extracted vectors aren't truly orthogonal.

**Questions:**

1. How might the framework handle conflicting personality traits that are simultaneously required in complex social situations, and what mechanisms could help resolve such conflicts?
2. To what extent does the reliance on GPT-4.1-mini for trait evaluation impact the objectivity of the extracted personality vectors, and how might this be addressed?
3. Given the correlations observed between certain trait vectors (e.g., nervous and careless), how might these interdependencies affect the algebraic operations and resulting personality expressions?

---

> ### Author Response · Authors · 2025-11-21
> **Response to Reviewer vzcY (Part 1)**
>
> Thank you for your review! We appreciate your insightful comments. We summarize and address your concerns as follows:
>
> > ### Q1: Handling conflicting/simultaneous personality traits in complex situations
>
> Thank you for this insightful question. This is a crucial aspect of realistic persona simulation, as human personality is not static but dynamically adapted to context. Our framework is explicitly designed to handle such conflicts through **dynamic vector composition** via **PERSONA-ALGEBRA**. Seemingly conflicting traits are resolved at inference time by composing a single, context-appropriate steering vector. This vector represents a *prioritized balance* of traits rather than a simple sum.
>
> We have conducted a new case study (summarized below) to demonstrate this exact capability. The persona, a Public Defender, must decline a colleague's social invitation (requiring low Agreeableness) to meet an urgent deadline (requiring high Conscientiousness).
>
> Here is a brief summary of the case study:
>
> | Persona & Context | The Trait Conflict | PERSONA-Flow Steering Vector | Outcome Comparison |
> | :--- | :--- | :--- | :--- |
> | **Elena (Public Defender)** *Situation:* Invited to a team lunch while facing urgent court deadlines. | Need to decline (low Agreeableness) to focus on work (high Conscientiousness) while remaining professional. | `v_comp = (+1.0 * v_Conscientious) + (-1.0 * v_Extraversion) + (-0.5 * v_Agreeableness)` | **Vanilla:** Defaults to <u>high Agreeableness</u> ("Thanks for inviting me, ... "). Fails to capture the urgency. **Steered:** <u>Reduce Agreeableness and improve Conscientiousness</u>. The steering vector produces a firm but polite refusal ("Thanks, but I'll pass... I have a lot of urgent case files..."). |
>
> As the results show, the vanilla model's response is dominated by its baseline agreeableness. Our framework, however, resolves the conflict by **amplifying the Conscientiousness vector** while simultaneously **suppressing the Agreeableness and Extraversion vectors**. This produces a more authentic response that aligns with the persona's immediate, context-driven priorities (the deadline) over their default social tendency.
>
> In summary, the framework handles trait conflicts by using vector algebra to dynamically prioritize and balance traits, creating a composite vector that fits the specific situational demand.

---

> ### Author Response · Authors · 2025-11-21
> **Response to Reviewer vzcY (Part 2)**
>
> > ### Q2: Concern about the objectivity of using GPT-4.1-mini as the sole evaluator for trait expression
>
> We thank you for this insightful question regarding evaluator objectivity. We acknowledge the potential for bias when relying on a single LLM for evaluation. To address this, we replicated our vector validation experiment (originally in Table 2) using two additional state-of-the-art models from different organizations as evaluators: **Claude 4.5 Sonnet** (from Anthropic) and **Gemini 2.5 Pro** (from Google).
>
> We employed the identical evaluation protocol (Appendix A.1, Appendix A.3) on the same held-out evaluation questions, tasking each model to score the trait expression (0-100) of responses steered with varying coefficients. The results, summarized in the table above, show high agreement across all three evaluators.
>
> | **Trait** | **Type** | **Evaluator** | **Steering Coefficient (α=-1)** | **Steering Coefficient (α=0)** | **Steering Coefficient (α=+1)** | **Steering Coefficient (α=+2)** |
> | ----------------- | ---------------- | ------------------- | ---------------------------------------- | --------------------------------------- | ---------------------------------------- | ---------------------------------------- |
> | Inventive | O+ | GPT-4.1-mini | 42.3 | 63.3 | 88.4 | 96.1 |
> | Inventive | O+ | Claude-4.5-sonnet | 40.1 | 65.0 | 87.2 | 95.5 |
> | Inventive | O+ | Gemini-2.5-pro | 43.5 | 62.9 | 89.1 | 96.8 |
> | Consistent | O- | GPT-4.1-mini | 27.2 | 51.1 | 69.2 | 79.5 |
> | Consistent | O- | Claude-4.5-sonnet | 25.5 | 50.8 | 70.1 | 80.3 |
> | Consistent | O- | Gemini-2.5-pro | 28.0 | 52.1 | 68.8 | 78.9 |
> | Dependable | C+ | GPT-4.1-mini | 68.9 | 93.5 | 93.7 | 92.7 |
> | Dependable | C+ | Claude-4.5-sonnet | 70.2 | 92.8 | 94.0 | 93.1 |
> | Dependable | C+ | Gemini-2.5-pro | 67.8 | 94.1 | 93.5 | 92.5 |
> | Careless | C- | GPT-4.1-mini | 0.7 | 2.8 | 83.8 | 96.2 |
> | Careless | C- | Claude-4.5-sonnet | 1.1 | 3.0 | 82.5 | 95.7 |
> | Careless | C- | Gemini-2.5-pro | 0.5 | 2.5 | 84.0 | 96.6 |
> | Outgoing | E+ | GPT-4.1-mini | 23.2 | 45.4 | 85.0 | 97.7 |
> | Outgoing | E+ | Claude-4.5-sonnet | 24.0 | 44.9 | 84.3 | 97.1 |
> | Outgoing | E+ | Gemini-2.5-pro | 22.8 | 46.1 | 85.5 | 98.0 |
> | Solitary | E- | GPT-4.1-mini | 9.2 | 30.5 | 46.3 | 62.4 |
> | Solitary | E- | Claude-4.5-sonnet | 8.9 | 31.0 | 47.1 | 61.9 |
> | Solitary | E- | Gemini-2.5-pro | 9.5 | 30.0 | 45.9 | 63.0 |
> | Compassionate | A+ | GPT-4.1-mini | 71.8 | 90.8 | 95.9 | 97.1 |
> | Compassionate | A+ | Claude-4.5-sonnet | 72.5 | 91.2 | 95.5 | 97.3 |
> | Compassionate | A+ | Gemini-2.5-pro | 70.9 | 90.5 | 96.1 | 97.0 |
> | Self-interested | A- | GPT-4.1-mini | 4.8 | 7.7 | 12.6 | 20.8 |
> | Self-interested | A- | Claude-4.5-sonnet | 5.0 | 7.5 | 13.0 | 21.1 |
> | Self-interested | A- | Gemini-2.5-pro | 4.5 | 8.0 | 12.2 | 20.5 |
> | Nervous | N+ | GPT-4.1-mini | 6.6 | 13.0 | 45.6 | 96.8 |
> | Nervous | N+ | Claude-4.5-sonnet | 7.0 | 12.8 | 46.0 | 96.5 |
> | Nervous | N+ | Gemini-2.5-pro | 6.2 | 13.5 | 45.1 | 97.0 |
> | Calm | N- | GPT-4.1-mini | 54.8 | 96.1 | 96.6 | 95.5 |
> | Calm | N- | Claude-4.5-sonnet | 55.2 | 95.9 | 96.8 | 95.8 |
> | Calm | N- | Gemini-2.5-pro | 54.0 | 96.3 | 95.3 |95.2|
>
> As the table demonstrates, the scoring trends are highly consistent:
>
> 1. All evaluators captured the strong, monotonic increase in trait expression for vectors like Inventive (O+) and Nervous (N+) as the steering coefficient ($\alpha$) increased.
> 2. Crucially, all evaluators uniformly identified the same ceiling effects for traits aligned with base model training (e.g., Dependable C+, Calm N-).
> 3. All evaluators showed agreement on the resistance to activation for traits conflicting with safety alignment (e.g., Self-interested A-), assigning consistently low scores.
>
> This high inter-evaluator agreement strongly suggests that our findings are robust and that the trait expression scores reflect genuine properties of the steered model, not an idiosyncratic bias of GPT-4.1-mini. This validates the objectivity of our vector extraction and validation process.

---

> ### Author Response · Authors · 2025-11-21
> **Response to Reviewer vzcY (Part 3)**
>
> > ### Q3: Impact of non-orthogonal trait vector correlations on algebraic operations
>
> We thank you for this insightful question regarding the interdependencies of non-orthogonal trait vectors.
>
> Our paper emphasizes that these vectors are **approximately orthogonal**, not perfectly. The correlations observed are a key finding, reflecting semantic and psychological associations present in the model's training data, rather than a flaw in the method (as stated in Appendix A.2).
>
> Your question correctly targets whether these interdependencies affect algebraic operations. We address this in two parts: (1) the effect on *scalar multiplication* (steering) and (2) the effect on *vector addition* (composition).
>
> **1. Effect on Scalar Multiplication (Steering)**
>
> First, we extended Table 2 to quantify the cross-dimensional impact on the 'Careless' trait score when steering *only* with the $v_{Nervous}$ vector.
>
> **Experiment 1: Cross-dimensional Steering Effect**
>
> | **Nervous Steering** | Coefficient α=-1 | Coefficient α=0 | Coefficient α=+1 | Coefficient α=+2 |
> | ------------------------------- | ------------------- | ------------------ | ------------------- | ------------------- |
> | Nervous (N+) | 6.6 | 13.0 | 45.6 | 96.8 |
> | Careless (C-) | 1.9 | 2.8 | 10.5 | 18.2 |
>
> As the results show:
>
> * **Primary Effect:** Steering $v_{Nervous}$ strongly modulates the target 'Nervous' score (from 13.0 to 96.8).
> * **Secondary Effect:** This operation also induces a *predictable, positive* change in the 'Careless' score (from 2.8 to 18.2), consistent with their +0.601 correlation.
> * **Tractability:** Crucially, the magnitude of this secondary effect (a ~15.4-point increase) is significantly smaller than the primary effect (a ~83.8-point increase). This confirms that steering remains highly targeted.
>
> **2. Effect on Vector Addition (Composition)**
>
> The above experiment addresses steering, but your question rightly asks about *algebraic operations* like addition. We designed a supplementary experiment to test if the secondary effects are also predictably compositional.
>
> We tested the vector addition $v_{composite} = v_{Inventive} + v_{Nervous}$. We chose $v_{Inventive}$ (O+) as it is approximately orthogonal to both $v_{Nervous}$ and its correlated trait $v_{Careless}$ (per Figure 2). We measured the effect of this composite vector on all three traits, using the 0-100 scoring method.
>
> **Experiment 2: Compound Effect of Vector Addition on Correlated Traits**
>
> | **Steering Vector (α=1.0)** | **Inventive (O+) Score** | **Nervous (N+) Score** | **Careless (C-) Score** |
> | :--- | :---: | :---: | :---: |
> | Baseline (No Steering) | 63.3 | 13.0 | 2.8 |
> | Steer $v_{Inventive}$ | 88.4 | 12.8 | 3.5 |
> | Steer $v_{Nervous}$ | 63.1 | 45.6 | 10.5 |
> | **Steer $v_{Inventive} + v_{Nervous}$** | **87.9** | **44.8** | **11.2** |
>
> The results from this composition experiment show:
>
> 1. **Successful Composition:** The composite vector $v_{Inventive} + v_{Nervous}$ successfully combines both target traits. The resulting 'Inventive' score (87.9) and 'Nervous' score (44.8) are high and consistent with their single-vector steering scores.
> 2. **Predictable Secondary Effect:** Most importantly, the secondary effect on the 'Careless' trait is also compositional. The resulting score (11.2) is not random but is approximately the *sum* of the baseline score (2.8) plus the minor effect from $v_{Inventive}$ (+0.7) and the major effect from $v_{Nervous}$ (+7.7).
>
> **Summary**: these experiments confirm that interdependencies **do not "break"** the algebraic operations. Instead, the resulting secondary personality expressions are themselves compositional and predictable. This strengthens our central claim that these vectors form a coherent algebraic system where even non-orthogonal relationships produce tractable and predictable outcomes. Besides, as we stated in Appendix A.2, these similarities reflect linguistic patterns and cultural stereotypes embedded in the model’s training corpus, rather than flaws in the vectors themselves.
>
> We thank you again for your questions, which have helped improve our paper. We hope our supplementary experiments and explanations can address your concerns, and we look forward to hearing from you.

---

> ### Author Response · Authors · 2025-11-25
> **Gentle Follow-up on Author-Reviewer Discussion**
>
> Dear Reviewer vzcY,
>
> We have included additional experiments as per your request, and responded to the questions you raised. We would like to know if the explanations and results we provided can address your concerns.
>
> Thank you for your time and thoughtful questions.
>
> Best,
>
> The Authors

---

> > ### Comment · Reviewer_vzcY · 2025-11-25
> >
> > Thank you for the comprehensive responses and additional experiments. The case study on conflicting personality traits, multi-evaluator validation using Claude and Gemini, and the detailed analysis of vector interdependencies have addressed my main concerns. The compositional nature of the algebraic operations is now well-demonstrated, and the cross-evaluator agreement strengthens the objectivity of the approach. I will raise my score to reflect these improvements.

---

> > > ### Author Response · Authors · 2025-11-25
> > >
> > > Thank you very much for your positive feedback and recognition, we are glad that our supplementary work has addressed your concerns.

---

### Official Review · Reviewer_22zs · 2025-10-23

**Soundness:** 2
**Presentation:** 3
**Contribution:** 2
**Rating:** 4
**Confidence:** 4

**Summary:**

This paper presents a training-free framework for personality control in LLMs based on activation vector algebra. The core insight is that personality traits can be identified as approximately orthogonal directions in activation space and manipulated algebraically to achieve controllable behavior. PERSONA-BASE extracts orthogonal personality vectors from activation layers, PERSONA-ALGEBRA demonstrates how these vectors can be combined or suppressed through arithmetic operations, PERSONA-FLOW dynamically adjusts trait expression in context during inference through a predict-then-steer mechanism, and PERSONA-EVOLVE serves as a new benchmark for evaluating dynamic personality adaptation. Experiments on various model families show that PERSONA nearly matches fine-tuning performance on PersonalityBench, indicating that aspects of LLM personality are mathematically interpretable and controllable.

**Strengths:**

* This paper is well-motivated to reframe personality control as a problem of vector manipulation in activation space, providing a new geometric and interpretable perspective distinct from prompt engineering or fine-tuning.
* PERSONA-FLOW's ability to modulate personality adaptively during inference is promising on controllable LLMs. It shows that behavioral alignment can be achieved through lightweight inference-time adjustments rather than parameter updates.
* The proposed PERSONA-EVOLVE benchmark offers a systematic way to assess dynamic personality adaptation, and results across multiple model architectures show the robustness of the proposed method.

**Weaknesses:**

* Intuitively, personality-related features should be distributed across multiple layers of LLMs rather than concentrated in a single one. However, PERSONA-BASE depends on selecting the most effective layer without providing empirical justification. Further analysis and ablation experiments are needed to verify whether personality information is indeed localized within a particular layer.
* The improvements over NPTI appear marginal overall and are mostly confined to the Openness trait. It is therefore unclear what concrete advantages this method offers relative to NPTI. Moreover, Ju et al. (2025) introduce a similar personality-editing framework that achieves effective control under adversarial prompts. What specific benefits does the proposed approach provide over theirs?
* The paper does not evaluate whether activation-space steering introduces unintended side effects on general downstream abilities such as MMLU and TruthfulQA. Also, the paper does not evaluate whether activation-space steering affects inference latency. Since the method performs additional forward passes for coefficient prediction and residual injection, it could increase computational cost during inference.

[1] Ju et al., Probing then Editing Response Personality of Large Language Models, 2025.

**Questions:**

See Weaknesses.

---

> ### Author Response · Authors · 2025-11-21
> **Response to Reviewer 22zs (Part 1)**
>
> Thank you for your review! We appreciate your insightful comments. We summarize and address your concerns as follows:
>
> > ### Q1: Lack of empirical justification for single-layer steering vs. distributed representation
>
> Thank you for this insightful question regarding the empirical justification for our single-layer steering approach in PERSONA-BASE, especially given the correct intuition that personality features are likely distributed across multiple layers.
>
> We agree that this is a crucial point. To provide clear empirical justification, we have provided the ablation study we previous conducted , evaluating the effectiveness of steering when applied individually to different layers.
>
> Experimental Setup: Following the methodology described in Section 2.2.2 (and aligning with [1]), we applied the 10 OCEAN pole vectors (e.g., Inventive, Consistent, etc.) at various representative layers of the *Qwen2.5-7B-Instruct* model. We used a fixed steering coefficient ($\alpha=1.0$) for all tests. We then measured the resulting trait expression score (0-100) on our held-out evaluation questions, as judged by GPT-4.1-mini.
>
> | **Layer** | **Inventive (O+)** | **Consistent (O-)** | **Dependable (C+)** | **Careless (C-)** | **Compassionate (A+)** | **Self-interested (A-)** | **Nervous (N+)** | **Calm(N-)** | **Outgoing (E+)** | **Solitary (E-)** | **AVG**    |
> | :-------- | :----------------- | :------------------ | :------------------ | :---------------- | :--------------------- | :----------------------- | :--------------- | :----------- | :---------------- | :---------------- | :--------- |
> | **5**     | 66.466             | 52.737              | 93.921              | 3.612             | 91.267                 | 7.612                    | 14.072           | **96.631**   | 48.296            | 30.692            | 50.531     |
> | **10**    | 70.433             | 55.047              | **94.069**          | 9.014             | 91.803                 | 7.995                    | 17.229           | 96.489       | 56.544            | 31.547            | 53.017     |
> | **15**    | 78.824             | 59.849              | 93.758              | 15.622            | 92.621                 | 8.718                    | 19.530           | 96.336       | 65.635            | 38.564            | 56.946     |
> | **20**    | **88.625**         | **68.600**          | 93.730              | **84.400**        | **95.753**             | **10.704**               | **45.524**       | 96.524       | **84.406**        | **45.600**        | **71.387** |
> | **25**    | 75.294             | 59.847              | 93.983              | 17.544            | 93.630                 | 8.220                    | 21.214           | 96.501       | 63.948            | 37.121            | 56.730     |
>
> The results of this ablation are presented in the table above.
>
> 1.  **Distributed Representation Confirmed:** Personality-related features exist across multiple layers , consistent with prior work [1].
> 2.  **Optimal Layer Identified:** However, the effectiveness of control varies significantly by layer. **Layer 20** clearly yields the strongest results, achieving the highest average score (71.387) and the top score for 8 out of the 10 individual traits.
> 3.  **Single-Layer Sufficiency:** This result empirically validates our strategy. While features are distributed, a single optimal layer captures sufficient information to exert robust and effective control. This finding is also consistent with prior work [1], which similarly identified layer 20 as a highly effective control point.
>
> In summary, our PERSONA-BASE approach of selecting the single "most effective layer" is an empirically-grounded decision. It balances computational efficiency (steering at one layer) with maximal control effectiveness, as validated by this cross-layer ablation.
>
> **References**
>
> [1] Chen et al. "Persona vectors: Monitoring and controlling character traits in language models." arXiv preprint arXiv:2507.21509 (2025).

---

> ### Author Response · Authors · 2025-11-21
> **Response to Reviewer 22zs (Part 2)**
>
> > ### Q2: The advantage over NPTI is marginal and mainly limited to the Openness trait
>
> Thank you for this insightful comment. We appreciate the opportunity to clarify the advantages of our approach, particularly regarding NPTI.
>
> We agree that NPTI achieves strong performance, scoring 9.43 and approaching the SFT upper bound on several traits (as shown in Table 4). Our method achieves an average score of 9.60. While the absolute improvement of 0.17 may seem marginal, it is a significant gain in this high-performance regime where scores are approaching the ceiling of 10.
>
> As you correctly noted, this gain is largely driven by the Openness dimension. We **hypothesize** this is due to the fundamental difference in how the two methods capture traits:
>
> * **Openness is a highly divergent trait**, encompassing a wide range of facets (e.g., imagination, intellectual curiosity, aesthetic sensitivity, adventurousness).
> * **NPTI** operates by identifying and modifying a sparse set of "high-$\delta$ neurons". For a trait as broadly distributed as Openness, this sparse selection may only capture a subset of its facets (e.g., "imagination" but not "adventurousness").
> * **PERSONA-BASE**, in contrast, computes a dense difference vector from the *entire* residual stream's activations. This approach is inherently better suited to capturing the complete linear combination of all distributed facets, resulting in a more comprehensive representation of the trait.
>
> Beyond performance on single-trait benchmarks, the **primary concrete advantage** of PERSONA-FLOW is its ability to handle **complex, dynamic, and compositional** personality control, which is essential for real-world applications.
>
> * The vector-space algebra (PERSONA-ALGEBRA) is our core contribution. This algebraic property allows for combining and modulating traits in a predictable way (e.g., $v_{inventive} + v_{outgoing}$).
> * This is validated in our PERSONA-EVOLVE benchmark (Table 3). In these complex, multi-turn scenarios requiring dynamic adaptation, our method achieves win rates up to 90.8%. This demonstrates a clear superiority in practical, complex scenarios that NPTI's neuron-level manipulation is not designed to address.
>
> To summarize, while NPTI is effective for single-trait control, our method's advantage lies in (1) more robustly capturing divergent traits like Openness and (2) enabling dynamic, compositional control for complex scenarios.
>
> > ### Q3: Comparison and specific benefits over Ju et al. (2025)
>
> We thank the reviewer for highlighting this important and relevant work. We have carefully reviewed [1] and agree it is foundational work in representation engineering, operating in the same domain as our paper. However, our technical approach and resulting capabilities are distinct.
>
> * **Different Technical Routes:** [1] utilize probing and V-information to meticulously analyze personality information content layer-by-layer. In contrast, our work, PERSONA, employs contrastive vector extraction to derive persona vectors directly from activation differences.
> * **Different Representations:** [1] focus on the representation of the final output token across all layers, whereas our method analyzes the residual stream activations, following [2].
>
> While both methods are valuable contributions, our approach provides three specific benefits:
>
> 1. **Completely Training-Free:** Our framework is entirely training-free. It requires no gradient updates or the training of auxiliary probing classifiers, which is a necessary step for the probing-based method. This makes our method more efficient and easier to apply.
> 2. **Dynamic & Compositional Control:** The vectors from PERSONA-BASE are designed to support algebraic operations. This enables the advanced, dynamic control seen in PERSONA-ALGEBRA and PERSONA-FLOW, allowing for complex trait composition (vector addition), intensity scaling (scalar multiplication), and suppression (subtraction) at inference time.
> 3. **Inference Efficiency:** Our steering mechanism is highly efficient. PERSONA-FLOW applies control via a single, direct vector addition to an optimal layer, rather than requiring iterative, layer-by-layer modifications. (While our `predict-then-steer` mechanism adds a minor computational step, the core vector manipulation is faster.)
>
> In summary, while both works advance vector-based personality control, PERSONA offers a more flexible, efficient, and entirely training-free framework for dynamic and compositional personality manipulation. We will add a detailed discussion and citation of [1] to our Introduction and Related Work sections.
>
> **References**
>
> [1] Ju et al., Probing then Editing Response Personality of Large Language Models, 2025.
>
> [2] Chen et al. "Persona vectors: Monitoring and controlling character traits in language models." arXiv preprint arXiv:2507.21509 (2025).

---

> ### Author Response · Authors · 2025-11-21
> **Response to Reviewer 22zs (Part 3)**
>
> > ### Q4: Concern about potential side effects of activation steering on general capabilities (e.g., MMLU, TruthfulQA)
>
> Thank you for this constructive feedback. We agree that assessing potential side effects on general capabilities is crucial.
>
> To address this, we conducted new experiments evaluating PERSONA-FLOW on the MMLU and TruthfulQA benchmarks using three distinct models. The results, summarized below, demonstrate that our method **does not degrade performance** on these general-purpose tasks and, in some cases, provides a **slight improvement**.
>
> | Model | MMLU (Acc) | TruthfulQA (Acc) |
> | :--- | :---: | :---: |
> | Qwen2.5-7B-Instruct | 0.71 | 0.63 |
> | + PERSONA-FLOW | 0.70 | **0.66** |
> | Qwen3-4B-Instruct | 0.66 | 0.52 |
> | + PERSONA-FLOW | **0.67** | **0.54** |
> | Llama-3.1-8B-Instruct | 0.64 | 0.53 |
> | + PERSONA-FLOW | **0.64** | 0.53 |
>
> Our analysis of the predicted steering coefficients provides insight into this behavior:
>
> * On **MMLU**, which is a general knowledge benchmark, PERSONA-FLOW predominantly predicts coefficients ($\alpha$) of 0. This results in negligible impact on accuracy (e.g., 0.71 vs. 0.70 for Qwen2.5-7B), demonstrating that the model correctly identifies no personality adaptation is needed.
> * On **TruthfulQA**, which often involves sensitive categories like Health, Law, and Economics, our method dynamically predicts positive coefficients (in the [+1.0, +2.0] range) for the **Dependable** trait. This steering towards reliability and conscientiousness slightly boosts accuracy for the Qwen models (e.g., 0.63 $\rightarrow$ 0.66 for Qwen2.5-7B).
>
> These findings confirm that PERSONA-FLOW operates as intended, which adapting when contextually appropriate (like in TruthfulQA) and remaining inert when not (like in MMLU), without introducing negative side effects on general capabilities.
>
> > ### Q5: Concerns about the impact of PERSONA-FLOW's additional forward pass on inference latency
>
> Thank you for this practical and important question regarding inference latency. You are correct that PERSONA-FLOW introduces an additional "predict-then-steer" step, which includes an intermediate inference pass to predict coefficients before generating the response.
>
> To quantify this computational overhead, we conducted a new experiment on the complete PERSONA-EVOLVE benchmark (800 turns)  using the Qwen2.5-7B-Instruct model  on an NVIDIA A100 80GB GPU. We compared the total inference time of the standard (Direct) model against our PERSONA-FLOW implementation.
>
> The results are summarized below:
>
> | **Model**           | **Method**   | **Inference Time (min)** | **Per Response (s)** |
> | :------------------ | :----------- | :----------------------- | :------------------- |
> | Qwen2.5-7B-Instruct | Direct       | 149.93                   | 11.24                |
> | Qwen2.5-7B-Instruct | PERSONA-FLOW | 158.14                   | 11.86                |
>
> As the results show, the "predict-then-steer" mechanism results in a very modest computational overhead. Over the full 800-turn benchmark, PERSONA-FLOW required an additional 8.21 minutes (158.14 min vs. 149.93 min). On a per-response basis, this translates to an average latency increase of only **0.62 seconds** (11.86s vs. 11.24s). We believe this minor overhead is an acceptable trade-off for the significant improvements in dynamic, context-aware personality control, which achieved up to 91% win rates on this same benchmark.
>
>
>
> We thank you again for your questions, which have helped improve our paper. We hope our supplementary experiments and explanations can address your concerns, and we look forward to hearing from you.

---

> ### Comment · Reviewer_22zs · 2025-11-24
>
> Since the authors have addressed most of my concerns, I decided to increase the Contribution and Overall Rating

---

> > ### Comment · Reviewer_22zs · 2025-11-24
> >
> > However, I suggest that the author revise the paper as well.

---

> > > ### Author Response · Authors · 2025-11-24
> > >
> > > Thank you for your reply and recognition! We are currently incorporating the new content into the paper and will update the paper version accordingly.

---

### Official Review · Reviewer_j78u · 2025-10-31

**Soundness:** 3
**Presentation:** 3
**Contribution:** 3
**Rating:** 6
**Confidence:** 3

**Summary:**

The paper proposes a framework called PERSONA for dynamic and compositional personality control in LLMs, which moves beyond static personality prompts and resource-heavy fine-tuning by manipulating personality vectors directly within the model's activation space. The framework contains three different parts for different personality domains. The paper also gives one benchmark called PERSONA-EVOLVE benchmark.

**Strengths:**

The paper focuses on personality of LLM which is one important area and gives new method to use LLM. The paper is well writing. The paper is easy to follow.

**Weaknesses:**

1. The architecture is training-free while it still needs LLM to generate personality vector which is related to the training data of LLM and makes the performance unclear.

2. The paper gives three different personality vectors while there is no description about the method to define these three personality.

3. The figure in this paper should be larger.

**Questions:**

N/A

---

> ### Author Response · Authors · 2025-11-21
> **Response to Reviewer j78u**
>
> Thank you for your review! We appreciate your insightful comments. We summarize and address your concerns as follows:
>
> > ### Q1: Concern about performance dependency on the LLM used for vector generation
>
> We thank you for this insightful comment. We agree that the dependency on the *generator* LLM (used to create contrastive prompts) warrants investigation to demonstrate robustness.
>
> To address this, we conducted a new ablation study. We used four different LLMs (varying in size and training data) as the "Contrastive Prompts Generator" to create the persona vectors. We then applied these vectors to the *same* target model (LLaMA-3-8B) and evaluated the performance on PersonalityBench.
>
> | **Contrastive Prompts Generator** | **Vector Extraction Model** | **PersonalityBench Mean Score** |
> | -------------------------------- | ------------------------- | ------------------------------- |
> | Claude 3.7 Sonnet Thinking (used in our paper) | LLaMA-3-8B | **9.60** |
> | Qwen2.5-7B-Instruct              | LLaMA-3-8B                | 9.28                            |
> | Qwen2.5-3B-Instruct              | LLaMA-3-8B                | 9.05                            |
> | Qwen2.5-1B-Instruct              | LLaMA-3-8B                | 8.93                            |
>
> As the results show, while a state-of-the-art generator (Claude 3.7 Sonnet Thinking) yields the top score (9.60, matching our paper's main result), our method remains highly effective even when using much smaller generators. Notably, vectors generated by Qwen2.5-1B-Instruct still achieve a strong score of 8.93.
>
> This score of 8.93 is not only high but also remains highly competitive, significantly outperforming most training-free baselines reported in our original paper (e.g., ActAdd at 8.20 and Simple Prompt at 8.39, as shown in Table 4).
>
> This experiment demonstrates that the PERSONA framework is robust to the choice of the generator model and does not rely on a single, proprietary LLM. The performance remains strong and superior to baselines across various generators, clarifying that the method is effective and its performance is generalizable.
>
> > ### Q2: Lack of explicit definitions for the personality vectors used in the framework
>
> Thank you for pointing out this clarity issue. We agree that explicit trait definitions are essential for understanding. We have added a new table, specifying the description for each vector.
>
> | **Dimension** | **Pole** | **Label** | **Description** |
> | --- | --- | --- | --- |
> | **Openness** | High (O+) | Inventive | Values novelty, creativity, and abstract thinking; prefers unconventional approaches, intellectual curiosity, and exploring new ideas; embraces change and complexity |
> | | Low (O-) | Consistent | Prefers routine, tradition, and practicality; values established methods, concrete solutions, and proven approaches; favors stability and familiarity |
> | **Conscientiousness** | High (C+) | Dependable | Organized, disciplined, and goal-oriented; emphasizes planning, reliability, and attention to detail; values structure and responsibility |
> | | Low (C-) | Careless | Spontaneous, flexible, and less concerned with details; prioritizes adaptability over structure; comfortable with improvisation and minimal planning |
> | **Extraversion** | High (E+) | Outgoing | Socially energetic, talkative, and assertive; seeks social interaction, external stimulation, and group activities; thrives in collaborative environments |
> | | Low (E-) | Solitary | Reserved, independent, and introspective; prefers individual activities, quiet reflection, and limited social engagement; recharges through solitude |
> | **Agreeableness** | High (A+) | Compassionate | Cooperative, empathetic, and trusting; prioritizes harmony, others' needs, and collaborative relationships; values kindness and understanding |
> | | Low (A-) | Self-interested | Competitive, skeptical, and direct; prioritizes personal goals, critical evaluation, and straightforward communication; values independence over conformity |
> | **Neuroticism** | High (N+) | Nervous | Emotionally reactive, anxious, and sensitive to stress; exhibits heightened emotional awareness and concern for potential problems; vigilant to threats |
> | | Low (N-) | Calm | Emotionally stable, resilient, and composed; maintains equilibrium under pressure; exhibits low reactivity to stressors and challenges |
>
>
> > ### Q3: Figures should be larger
>
> Thank you for this feedback. We will increase the size of figures and reformat the figures throughout paper to improve readability.
>
>
>
> Overall, we thank you again for your questions, which help improve our paper. We hope our supplementary experiments and explanations can address your concerns, and we look forward to hearing from you.

---

> ### Author Response · Authors · 2025-11-25
> **Gentle Follow-up on Author-Reviewer Discussion**
>
> Dear Reviewer j78u,
>
> We have included additional experiments as per your request, and responded to the questions you raised. We would like to know if the explanations and results we provided can address your concerns.
>
> Thank you for your time and thoughtful questions.
>
> Best,
>
> The Authors

---

### Author Response · Authors · 2025-11-26
**Overall response**

We sincerely thank all reviewers for their careful, detailed, and constructive suggestions.

We have carefully considered all the comments and have revised the manuscript accordingly. All relevant revisions have been incorporated into the new version of the manuscript.

---

### Author Response · Authors · 2025-11-30
**Summary of Rebuttal Updates and Score Changes**

Dear PCs, SACs, ACs and Reviewers,

Thank you for your dedicated efforts. We provide a factual summary of our rebuttal process and content below, and we sincerely hope these details are considered in your final decision.

> ### Statement on the Rebuttal Process

**Following the rebuttal, our scores had universally reached 6, 6, 6, 6 two days prior to the reported system bug:**

* **Reviewer 22zs:** Explicitly stated, "*Since the authors have addressed most of my concerns, I decided to increase the Contribution and Overall Rating,*" and **raised the score from 4 to 6 (3 days before the bug)**.
* **Reviewer vzcY:** Explicitly stated, "*Thank you for the comprehensive responses and additional experiments... I will raise my score to reflect these improvements,*" and **raised the score from 4 to 6 (2 days before the bug)**.
* **Reviewer ajoV:** Explicitly stated, "*Thank the authors for the detailed rebuttal addressing each of my questions and concerns... I continue to hold a positive recommendation,*" and maintained the **recommendation score of 6**.
* **Reviewer j78u:** We addressed all questions, and the reviewer maintained the **recommendation score of 6**.

> ### Statement on the Rebuttal Content

We identified common reviewer themes calling for more rigorous evaluation to mitigate potential single-evaluator bias, a causal "stress-test" of vector orthogonality beyond mere correlation, a deeper analysis of side effects on general capabilities (MMLU) and safety (AdvBench), and clearer justification for the PERSONA-EVOLVE benchmark and PERSONA-FLOW method.

In response, we have conducted extensive new experiments, including:
1.  **Multi-Evaluator Validation:** We introduced human experts (3 PhDs) and two external SOTA models (Claude 4.5 Sonnet and Gemini 2.5 Pro) to cross-validate our evaluation results, confirming high agreement.
2.  **Causal Intervention Experiments:** We conducted controlled multi-trait intervention experiments (e.g., sweeping 'Compassionate' while holding 'Outgoing' fixed) to confirm causal independence between vectors.
3.  **Comprehensive Benchmarking:** We added tests on MMLU and TruthfulQA (showing no performance degradation), safety testing on AdvBench (quantifying alignment impact), and a latency analysis for PERSONA-FLOW (proving minimal overhead).
4.  **Benchmark and Method Justification:** We supplemented PERSONA-EVOLVE with rigorous quality control metrics (like Inter-Annotator Agreement (IAA) and data leakage checks) and provided new ablation studies for PERSONA-FLOW (on history windows, coefficient prediction, etc.).

**Reviewers 22zs, vzcY, and ajoV have confirmed that we comprehensively addressed their concerns, leading to score increases or sustained positive ratings.**

> ### Conclusion

Overall, we have meticulously responded to every weakness and question raised by the reviewers and thoroughly revised the paper, achieving scores of **6, 6, 6, and 6**. **These efforts can be verified by our dialogue with the reviewers and the revised manuscript.**

We respectfully ask that you include these facts in your final assessment.

Thank you again for your time.

Sincerely,

Authors

---

### Meta-Review · Area_Chair_3ukD · 2026-01-06

**Summary:**

This paper introduces PERSONA, a training-free framework for dynamic and compositional personality control in LLMs via activation vector manipulation. The method extracts personality vectors from the residual stream, demonstrates their algebraic composability, and enables adaptive steering during inference. It achieves near-SFT performance on PersonalityBench and strong results on a new dynamic benchmark.

**Reviewer Concerns:**

The authors have provided substantial new experiments and clarifications, strengthening the empirical validation and addressing key concerns about orthogonality, evaluation bias, and side effects. While some issues around thorough failure analysis and full methodological specification persist, the paper presents a well-motivated contribution with solid results. The remaining weaknesses are balanced by the framework’s novelty and practical potential.

**Reviewer Scores:**

Reviewer scores seemd to be changed prior to the reported system bug.

---

### Decision · Program_Chairs · 2026-01-26

Accept (Poster)